# Fine-scale genomic analyses of admixed individuals reveal unrecognized genetic ancestry components in Argentina

Pierre Luisi[1]*, Angelina García[1,2,3], Juan Manuel Berros[4], Josefina M. B. Motti[5], Darío A. Demarchi[1,2,3], Emma Alfaro[6,7], Eliana Aquilano[8], Carina Argüelles[9,10], Sergio Avena[11,12], Graciela Bailliet[8], Julieta Beltramo[8,13], Claudio M. Bravi[8], Mariela Cuello[8], Cristina Dejean[11,12], José Edgardo Dipierri[7], Laura S. Jurado Medina[8], José Luis Lanata[14], Marina Muzzio[8], María Laura Parolin[15], Maia Pauro[1,2,3], Paula B. Paz Sepúlveda[8], Daniela Rodríguez Golpe[8], María Rita Santos[8], Marisol Schwab[8], Natalia Silvero[8], Jeremias Zubrzycki[16], Virginia Ramallo[17], Hernán Dopazo[18,19]*

1 Departamento de Antropología, Facultad de Filosofía y Humanidades, Universidad Nacional de Córdoba, Córdoba, Argentina, 2 Instituto de Antropología de Córdoba (IDACOR), Consejo Nacional de Investigaciones Científicas y Técnicas – Universidad Nacional de Córdoba, Córdoba, Argentina, 3 Universidad Nacional de Córdoba, Facultad de Filosofía y Humanidades, Museo de Antropología, Córdoba, Argentina, 4 Consejo Nacional de Investigaciones Científicas y Técnicas – Laboratorio de Análisis de Datos, Biocódices S.A., Buenos Aires, Argentina, 5 Núcleo de Estudios Interdisciplinarios de Poblaciones Humanas de Patagonia Austral (NEIPHA), Consejo Nacional de Investigaciones Científicas y Técnicas – Universidad Nacional del Centro de la Provincia de Buenos Aires, Quequén, Argentina, 6 Instituto de Ecorregiones Andinas (INECOA), Consejo Nacional de Investigaciones Científicas y Técnicas – Universidad Nacional de Jujuy, Jujuy, Argentina, 7 Instituto de Biología de la Altura, Universidad Nacional de Jujuy, Jujuy, Argentina, 8 Instituto Multidisciplinario de Biología Celular (IMBICE), Consejo Nacional de Investigaciones Científicas y Técnicas – Comisión de Investigaciones Científicas – Universidad Nacional de La Plata, La Plata, Argentina, 9 Departamento de Genética, Grupo de Investigación en Genética Aplicada (GIGA), Facultad de Ciencias Exactas, Químicas y Naturales, Instituto de Biología Subtropical (IBS)—Nodo Posadas, Universidad Nacional de Misiones (UNaM)–Consejo Nacional de Investigaciones Científicas y Técnicas, Posadas, Argentina, 10 Cátedra de Biología Molecular, Carrera de Medicina, Facultad de Ciencias de la Salud, Universidad Católica de las Misiones (UCAMI), Posadas, Argentina, 11 Instituto de Ciencias Antropológicas (ICA), Facultad de Filosofía y Letras, Universidad de Buenos Aires, Buenos Aires, Argentina, 12 Centro de Estudios Biomédicos, Biotecnológicos, Ambientales y Diagnóstico (CEBBAD), Universidad Maimónides, Buenos Aires, Argentina, 13 Laboratorio de Análisis Comparativo de ADN, Corte Suprema de Justicia de la Provincia de Buenos Aires, La Plata, Argentina, 14 Instituto de Investigaciones en Diversidad Cultural y Procesos de Cambio (IIDyPCa), Consejo Nacional de Investigaciones Científicas y Técnicas – Universidad Nacional de Río Negro, San Carlos de Bariloche, Argentina, 15 Instituto de Diversidad y Evolución Austral (IDEAus), Consejo Nacional de Investigaciones Científicas y Técnicas – Centro Nacional Patagónico, Puerto Madryn, Argentina, 16 Laboratorio de Genómica. Biocódices S.A., Buenos Aires, Argentina, 17 Instituto Patagónico de Ciencias Sociales y Humanas (IPCSH) – Consejo Nacional de Investigaciones Científicas y Técnicas—Centro Nacional Patagónico, Puerto Madryn, Argentina, 18 Departamento de Ecología, Genética y Evolución, Facultad de Ciencias Exactas y Naturales, Universidad de Buenos Aires, Buenos Aires, Argentina, 19 Consejo Nacional de Investigaciones Científicas y Técnicas – Biocodices S.A., Buenos Aires, Argentina

* pierrespc@gmail.com (PL); dopazoh@gmail.com (HD)



**Data Availability Statement:** The data analyzed here comprises both newly generated and previously reported data sets. Access to publicly available datasets should be requested through the

## Abstract

Similarly to other populations across the Americas, Argentinean populations trace back their genetic ancestry into African, European and Native American ancestors, reflecting a complex demographic history with multiple migration and admixture events in pre- and post-colonial times. However, little is known about the sub-continental origins of these three main ancestries. We present new high-throughput genotyping data for 87 admixed individuals

distribution channels indicated in each published study. Newly generated samples have been registered under study EGAS00001004492 in the European Genome-Phenome Archive which contains both raw and processed individual genotype datasets with accession number EGAD00010001913 and EGAD00001006227, respectively.

**Funding:** This project has been founded by FONCyT) - ANPCyT (grants PICT2014-1597 attributed to HD and PICT3655-2016 attributed to PL) and CONICET (grant PIP 0208/14 attributed to HD and VR). FONCyT: Fondo para la Investigación Científica. https://convocatoriasfoncyt.mincyt.gob. ar/ ANPCyT: Agencia Nacional de Promoción Científica y Tecnológica. https://www.argentina. gob.ar/ciencia/agencia CONICET: Consejo Nacional de Investigaciones Científicas y Técnicas. https:// www.conicet.gov.ar/ Biocódices provided support in the form of part-time employment salaries for authors HD, JMB and JZ. The funders had no role in study design, data collection and analysis, decision to publish, or preparation of the manuscript."

**Competing interests:** I have read the journal's policy and the authors of this manuscript have the following competing interests: PL provides consulting services to myDNAmap S.L. JMB and JZ are employed by Biocódices S.A. HD is the scientific director of Biocódices S.A. This does not alter our adherence to PLOS ONE policies on sharing data and materials.

**Abbreviations:** 1KGP, 1000 Genomes Project; AMBA, Metropolitan Area of Buenos Aires; BIC, Bayesian Information Criterion; CAN, Central Andes; CCP, Central Chile / Patagonia; CLM, Colombians from Medellin; CWA, Central Western Region in Argentina; CYA, Cuyo Region in Argentina; DS, Data set; LD, Linkage Disequilibrium; NEA, Northeastern Region in Argentina; NWA, Northwestern Region in Argentina; PCA, Principal Component Analysis; PEL, Peruvians from Lima; PPA, Pampean Region in Argentina; PTA, Patagonia Region in Argentina; SNP, Single Nucleotide Polymorphism; STF, Subtropical and Tropical Forests; YRI, Yoruba individuals in Ibadan, Nigeria.

across Argentina. This data was combined to previously published data for admixed individuals in the region and then compared to different reference panels specifically built to perform population structure analyses at a sub-continental level. Concerning the Native American ancestry, we could identify four Native American components segregating in modern Argentinean populations. Three of them are also found in modern South American populations and are specifically represented in Central Andes, Central Chile/Patagonia, and Subtropical and Tropical Forests geographic areas. The fourth component might be specific to the Central Western region of Argentina, and it is not well represented in any genomic data from the literature. As for the European and African ancestries, we confirmed previous results about origins from Southern Europe, Western and Central Western Africa, and we provide evidences for the presence of Northern European and Eastern African ancestries.

## Introduction

The first systematic investigation of human genetic variation in Argentina focused on a limited number of markers either uniparental (mtDNA, Y-STRs, Y-SNP; [1–10] or autosomal (Short Tandem Repeats, Ancestry Informative Markers, Alu sequences, indels, and blood groups [11–16]). Studies based on autosomal markers identified an important inter-individual heterogeneity for the African, Native American and European genetic ancestry proportions [11–16]. Accordingly, most of the studies based on uniparental markers showed large differences in the genetic composition of Argentinean populations, accounting for the different demographic histories within the country [1,2,17–20,3–9,16]. Although the idea of a 'white' country with most of inhabitant's descendants from European immigrants has now been rejected by these studies, the Argentine founding myth of a white and European nation remains perceptible today [21].

There is a wealth of information about the European ancestors, both from familial stories and from historical records. On the contrary, little is known about the African and Native American populations that contributed to the admixture events between the continental components. The great wave of European immigration occurred between the mid-19th to mid-20th centuries. Although immigrants came from all over the continent, historical records attest that immigration waves from Southern Europe (mainly Italy and Spain) were predominant [22].

As for the African genetic origins in Argentina, historical records about the arrival of African slaves to to Río de la Plata show that Luanda, in current Angola territory in Central Western region of Africa, was the main departure harbor, followed by ports located in the Gulf of Guinean and on the coast of the present-day Senegalese, Gambian and Sierra Leone territories. The Indian Ocean coasts, in the current territory of Mozambique, were also an important departure location [23–25]. Although the departure harbor is a poor proxy to infer the actual slaves' origins [25,26], the African uniparental lineages observed in Argentina are consistent with the historical records [27,28].

As for the Native American component, it is difficult to study its origin focusing on present-day communities since their organization has changed drastically after the arrival of the first conquerors in the 16th century [21]. During the period of conquest and colonization, wars, diseases and forced labor decimated the Native populations. The system of colonial exploitation also often meant the relocation of individuals, families, and communities [29]. Then, the expansion of the nation-state by the late 19th Century can be described as a

territorial annexation process and subjugation of the indigenous peoples perpetrated by the Argentinean national armed forces between 1876 and 1917 [30].

Due to the specificity of the Argentinean demographic history, a remaining challenge is to unravel which populations from each continent contributed to the genetic pool in nowadays Argentinean populations leveraging genotype data for hundreds of thousands of autosomal markers from the whole genome. Recently, two articles presented high-throughput genotyping data for modern Argentine individuals, and provided the first insights to decipher which populations from each continent contributed to the genetic pool in nowadays Argentinean populations [31,32]. In both studies, it was found that the European ancestry in Argentina is mainly explained by Italian and Iberian ancestry components. Homburger *et al.* observed a strong gradient in Native American ancestry of South American Latinos between Andean and other South American Native American populations [31]. In addition, Muzzio *et al.* found that African ancestry is explained by a Central Western (Bantu-influenced) component and a Western component.

Despite these progresses, the efforts to understand the demographic history that shaped genetic diversity in Argentina have inevitably been scarce, particularly for African and Native American components. Studies of a wider region, namely South America can provide important insights. A previous study showed that Western and Central Western African ancestries are common across the Americas, particularly in Northern latitudes, while the influence of South/Eastern African ancestry is greater in South America [33]. In another study in Brazil, two African ancestries have been observed: a Western African one and another associated with Central East African and Bantu populations, the latter being more present in the Southeastern and Southern regions [34]. Ancient DNA analyses of the peopling of the Americas suggest that Native American populations of South America descend from two streams from Northern America: one mainly present in the Andes and another one present elsewhere [35]. These streams have replaced the first people settled in the region, whose ancestry was related to the Clovis culture [35]. Genetic continuity for the Native American component appears to have prevailed in the region ever since these replacements [35–37]. However, little is known about the legacy in modern populations of the different ancestries associated to the several waves of population arrivals in South America. Modern Native American populations in the Southern Cone of South America seems to be divided between a component that includes populations in Tropical and Subtropical Forests and another component that includes Andean, Central Chile and Patagonian populations [38]. A recent genomic study of ancient and modern populations from Central Chile and Western Patagonia further identified that they are differentiated from the Andean and Subtropical and Tropical Forests populations [37]. In another genomic study of modern samples (in which the Southern Cone is only represented by the Gran Chaco region), it has been found that all non-Andean South American populations are likely to share a common lineage, while they are unlikely to share with the Andeans any common ancestor from Central America [39], supporting the hypothesis of many back migrations to Central America from non-Andean South American populations [39]. It has also been described that the genetic interactions between the peopling routes on both sides of the Andes were limited [39,40].

Here, we carry out a fine-scale population genomic study to get insight into the genetic structure and the complex origins of the African, European and Native American ancestries in Argentina populations. Using Affymetrix, Axiom-Lat1 array technology, we genotyped 87 admixed individuals nationwide for more than 800,000 SNPs covering coding and non-coding regions of the genome. Additionally, a dataset with ~500 individuals from modern and ancient populations throughout South America was constructed. We also pulled together genotype data for Sub-Saharan African and European populations from the literature. The new data

generated in the present study, compared to those data sets, allows to broaden the knowledge of the sub-continental origins of the three main genetic ancestries of Argentinean populations.

## Results and discussion

### Studied populations

High-throughput genotyping data was generated for 87 admixed individuals throughout Argentina (**Fig 1** and **S1 Table**). For clarity in the visualization of the results, we used provinces and regions to classify the admixed individuals analyzed as shown in **Fig 1**. This classification has not been used for any statistical analyses. The generated data was compared to different data sets to understand the origins of the genetic diversity in the country. These data sets are called **DS<n>** and are described in **S2 Table**.

### Ancestry in a worldwide context

In order to characterize the genetic diversity observed in Argentina within a worldwide genetic context, Principal Component Analysis (PCA [41]; **S1 Fig**) was applied to the dataset **DS1**,

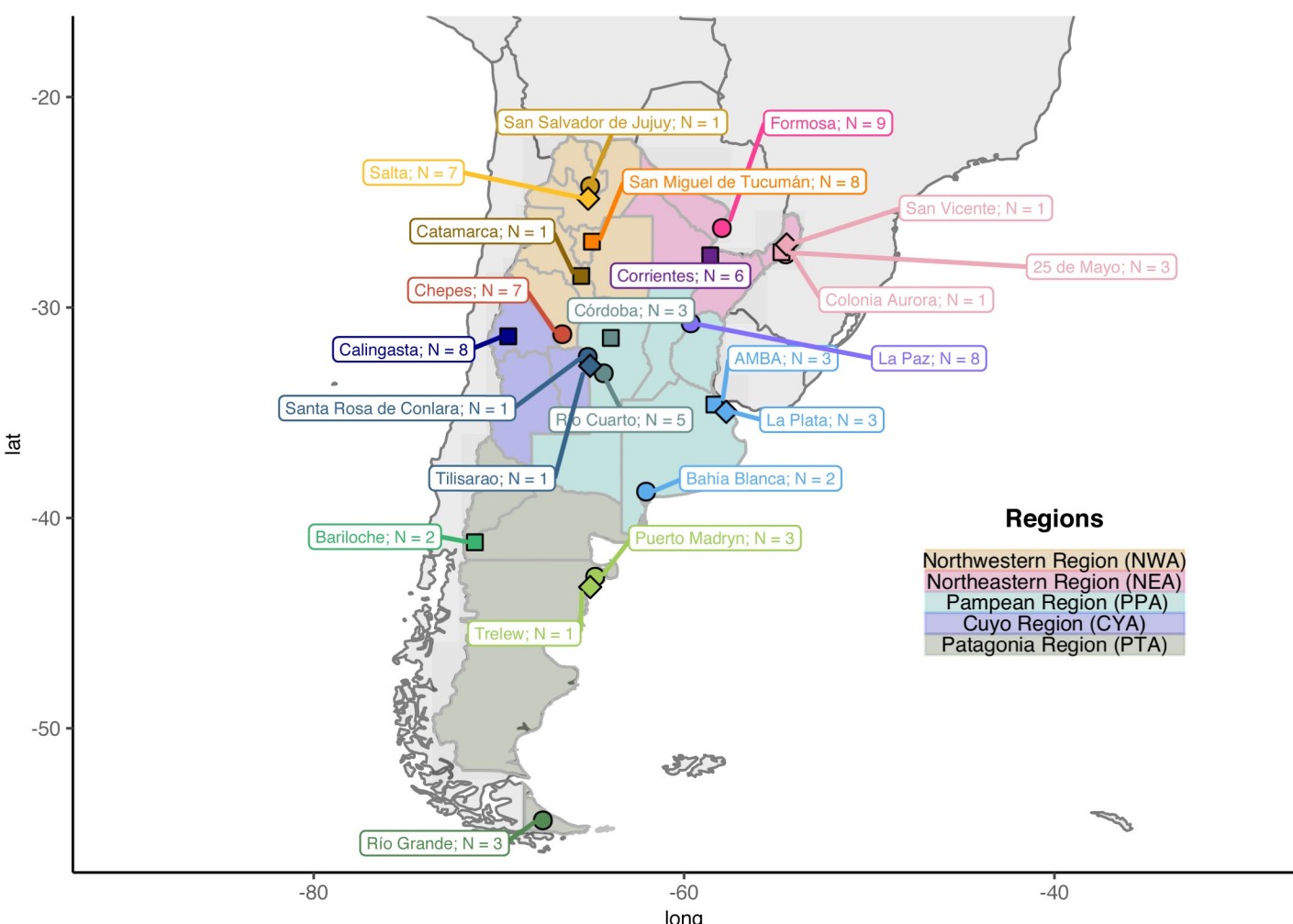

**Fig 1. Sample locations from the present study.** The samples are divided according to regions for visualization only. N: The number of samples that passed genotyping Quality Controls.

which contains genotype data for the samples from Argentina generated in the present study, as well as other admixed and Native American individuals from South America [31,37,38,42] and individuals from Europe and Africa from the 1000 Genomes Project (1KGP; [42]). The PCA shows that Argentinean individuals have different proportions of European, Native American and, much less represented, African ancestry (S1A Fig). This pattern, which has already been documented in other admixed populations from South America [31,42], including in Argentina [13–16,32] supports the heterogeneous genetic origins throughout the country. In addition, PC4 discriminates between Southern and Northern European individuals (S1B Fig) while PC6 separates Luhya population, a Bantu-speaking population in Kenya, from Western African populations (S1C Fig). The Argentinean samples do not exhibit any gradient along these two PCs. But, PC3 and PC5 discriminate three different main Native American ancestries that are also represented in the Argentinean samples.

We ran unsupervised clustering models with Admixture software [43] on **DS1**. We used from 3 to 12 putative ancestral populations. Comparing the cross-validation scores obtained for each run, we estimated that the genotype data analyzed was best explained with a model with $K = 8$ ancestral populations (S2A Fig). At $K = 3$ (S2B Fig and S3 Table), the algorithm allows estimating the proportions of European, Native American and African ancestry. For $K = 4$ to $K = 7$, (S2C–S2F Fig), the model detects sub-continental ancestries. At $K = 8$ (Fig 2; S3 Table), the European ancestry is divided into Northern and Southern components (dark

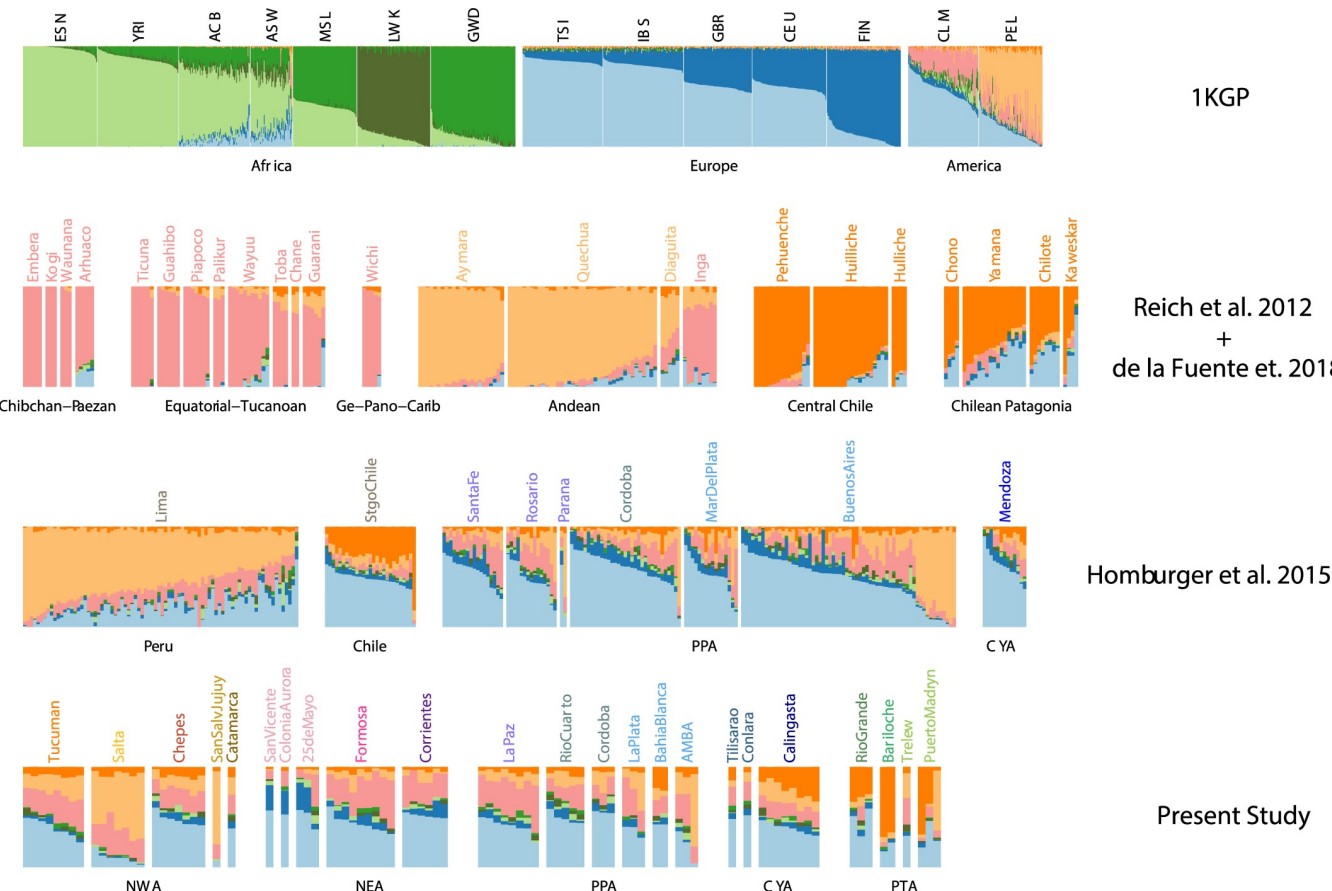

**Fig 2. Admixture analyses in a worldwide context.** Admixture for $K = 8$. **1st row**: 1000 Genomes Samples from Europe, Africa and South America [42]. **2nd row**: Modern Native American samples grouped following [38]. **3rd row**: Chilean, Peruvian and Argentinean admixed samples from [31]. **4th row:** Argentinean admixed samples from the present study. Samples are grouped according to regions (Fig 1); CYA: Cuyo Region in Argentina; NEA: Northeastern Region; NWA: Northwestern Region; PPA: Pampean Region; PTA: Patagonia Region. CLM: Colombians from Medellin; PEL: Peruvians from Lima.

and light blue, respectively), while African ancestry is composed of Westernmost African (green), Gulf of Guinea (light green) and Bantu-influenced (dark green) components. Moreover, three components are observed for Native American ancestry: one in Central Chile/Patagonia populations (in orange; hereafter referred to as **CCP**), another in Central Andes populations (in yellow; hereafter referred to as **CAN**) and a third one in populations from Tropical and Subtropical Forests (in pink; here after referred to as **STF**).

We confirmed that the estimates of the continental ancestry proportion obtained from a model with $K = 3$ and $K = 8$ are highly consistent (**S3 Fig**). Moreover, the eigenvectors from PCA and the ancestry proportion estimates with Admixture are well correlated (**S4 Fig**).

From the Admixture results for $K = 8$ (**Fig 2**), we observed that the European ancestry for Argentinean samples, is divided in Southern and Northern components, the former being the most abundant. The low proportion of African ancestry in Argentinean samples makes difficult to interpret its sub-continental origins from analyses within a global context. Surprisingly, all three components of Native American ancestry are present in most Argentinean samples (**Fig 2**). They exhibit mid proportions of different Native ancestries suggesting either the result of a mixture between these three ancestry components or an underrepresentation of Native American in the reference panels currently available. Such mixture pattern is not observed in other South American countries. Indeed, the Native American ancestry for Peruvian, Chilean and Colombian admixed samples is mainly represented by CAN, CCP and STF, respectively. This is consistent with the geographical area where the admixed individuals have been sampled, and the genetic ancestry of the indigenous communities from each country.

## Sub-continental ancestry components in Argentina

To decipher the sub-continental origins of the European, African and Native American ancestries, we first estimated local ancestry patterns in phased **DS2** and **DS3** (**DS2p** and **DS3p**) separately. Across phased chromosomes, we assigned whether a genomic region was of Native American, European or African ancestry using RFMix v2 [44]. We applied principal component and unsupervised clustering analyses on the masked data (**S5 Fig**) compiled with reference data sets describing the genetic diversity within each continent (**S2 Table**).

**The genetic legacy of European migration in Argentina.** We used **DS4**, a combination of the masked genotype data for admixed individuals. with a set of European individuals carefully selected to be representative of the genetic diversity in their sampling area [45,46].

From the PCA, we observed that most Argentinean individuals cluster with Iberians and Italians (**Fig 3**), as previously described [31,32]. However, some individuals cluster with Central and Northern Europeans.

We estimated that the genotype data analyzed was best explained with an admixture model with $K = 2$ (**S7A Fig**). Although it is rather difficult to assign a clear label to each of these two ancestral populations, it seems that the algorithm discriminates between Central/Northern (dark blue) and Southern Europe (light blue) components (**S7B Fig**). For $K = 3$, we still observe a Central/Northern European component (dark blue), while the South can be divided into a component particularly represented in Iberian individuals (turquoise) and another component more frequent in Southeastern Europeans and Italians (light blue). The Iberian component is highly represented in Southern American samples (**S7C Fig**), most likely reflecting the legacy of first Iberian migrations in South America during colonial period [11]. Samples from Argentina and Chile exhibit higher proportions of Southeastern/Italian and Northern European ancestries than Colombians, as well as lower Iberian ancestry proportions (**S8 Fig**). We observed no significant difference in the proportions of any European ancestry between Argentinean and Chilean samples (**S8 Fig**). However, both PCA and Admixture shows that

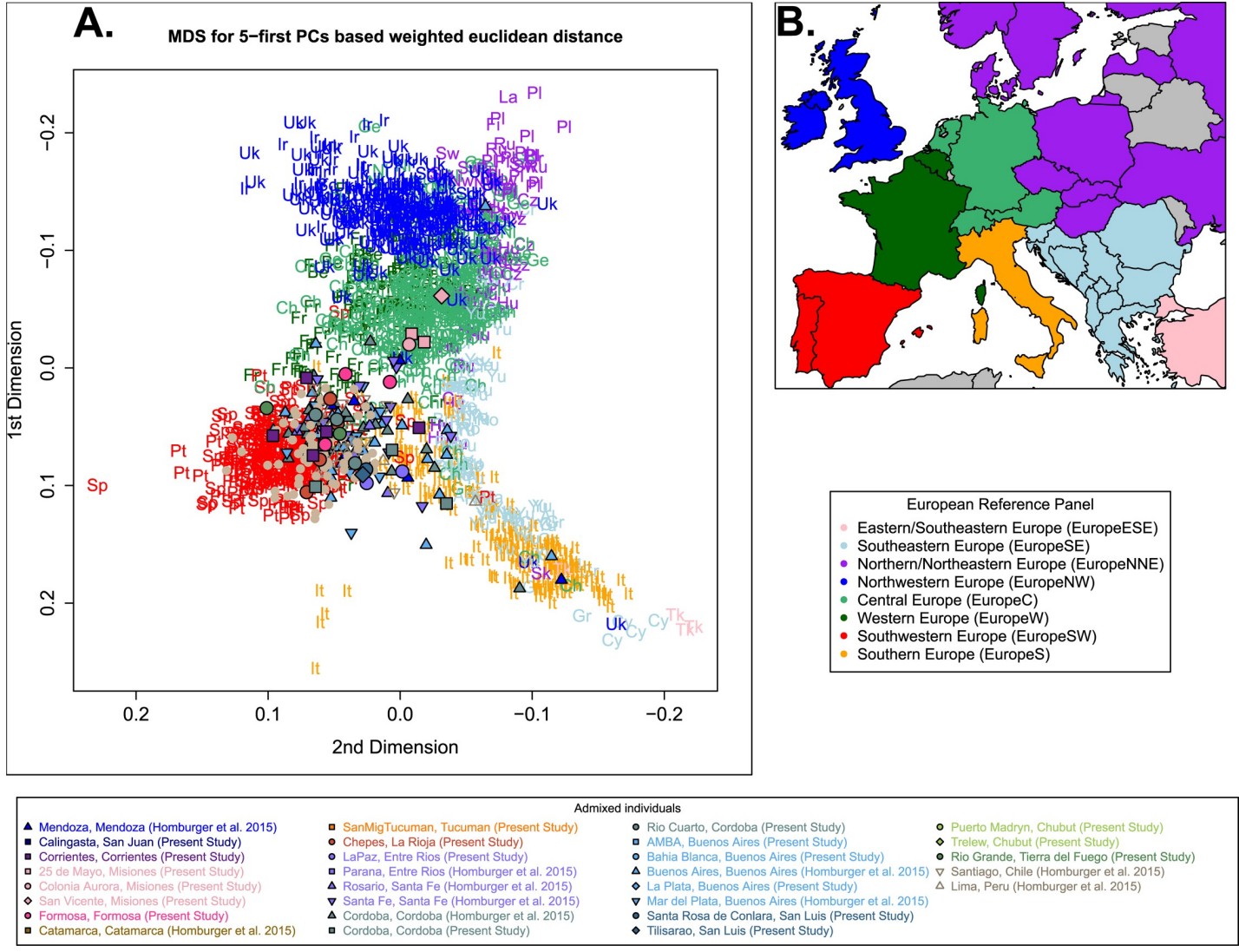

**Fig 3. European ancestry specific principal component analysis. (A)** Multidimensional scaling scatterplot (MDS) from Euclidian distances calculated from weighted 5-first European-specific Principal Components performed using the European reference samples, and admixed samples masked for African ancestry. Individuals from the European reference panel are colored according to main European geographic regions as shown in **(B)** while South American individuals are represented as shown in the legend. Elbow method to choose the best number of PCs to compute MDS is shown in **S6 Fig**.

the individuals with most Southeastern/Italian ancestry are from Argentina, This is consistent with a previous study [31], and it can be explained by the many arrivals from Italy during the great wave of European immigration in the 19th and 20th centuries [22].

Moreover, Argentinean individuals with higher proportion of Central/Northern European ancestry are from Misiones province (NEA), consistent with the historical record of settlement of Polish, German, Danish and Swedish colonies promoted by governmental or private enterprises in the province [47].

**Different African ancestry components in Argentina.** To investigate the sub-continental components that explains African ancestry in Argentina, we used **DS5** which combines masked genotype data from admixed individuals with a published data set of African individuals [42,48–50] (**S9A Fig**). The African reference populations used here can be divided into five

main groups: Bantu-influenced, Hunter-Gatherers, Western African, San and Eastern African populations [48].

From the PCA applied to this data, we observed that African ancestry in Argentina has small genetic affinity with San and Hunter-Gatherers populations. Indeed, according to PC1 and PC2, Argentinean individuals are located within a group of African individuals belonging to Bantu-influenced, Western African and Eastern African populations, and clearly distinct from San and Hunter-Gatherer populations (S9B Fig). Moreover, on PC3, most Argentinean are closer to Western African and Bantu-influenced populations, while several Argentinean individuals, particularly from Northern Argentina, are located within a gradient towards Eastern African populations (S9D Fig). According to PC4, most of the Argentinean individuals are closer to the Bantu-influenced populations while others individuals are closer to Western Africa populations, and several other Argentinean individuals are found in a cline between these two groups (S9C Fig). These patterns were confirmed by applying Admixture algorithm. The Cross-Validation procedure points to a best fit of the data with $K = 5$ (S10A Fig), a model for which we observe that he Bantu-influenced and Western Africa components are the most represented in Argentinean individuals. Moreover, this analysis also showed that some individuals exhibit smaller, yet important, proportions of Eastern African ancestry, particularly in Northern Argentina (Fig 4). Although, the important missing genotype rate in masked data for admixed individuals could bias PCA and Admixture results, the results obtained by both methods are highly consistent for admixed individuals (S11 Fig).

The legacy of Western Africa on the African genetic diversity in the Americas has been preeminent [51–53], along with an impact of Bantu-influenced populations from Central Western Africa, particularly in Brazil and the Caribbean [51,52]. These two African ancestries have also been previously documented in Argentina from studies of autosomal [32] and maternal markers [27,28]. Maternal lineages specific to populations from Central Western Africa–particularly from Angola- are the most common African lineages in Argentina according to studies in the Central region [27] and in four urban centers (Puerto Madryn, Rosario, Resistencia and Salta) [28]. These results are concordant with the predominance of the Bantu-influenced origin identified in the present study. In addition, the presence of Southeastern Africa maternal lineages

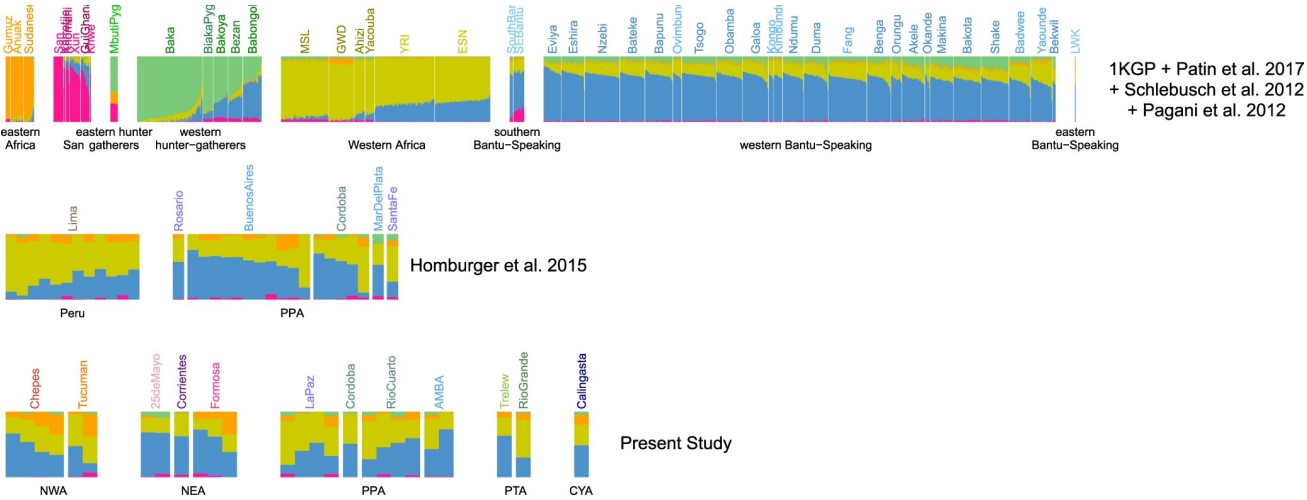

**Fig 4. African ancestry-specific admixture analysis.** Admixture for $K = 5$. **1st row**: 1,685 reference samples with >99% of African ancestry. Populations are grouped following [48]. **2nd and 3rd rows**: Peruvian and Argentinean admixed samples from [31] and from the present study. Samples are grouped according to region. CYA: Cuyo Region; NEA: Northeastern Region, NWA: Northwestern Region; PPA: Pampean Region; PTA: Patagonia Region. Genotype data for admixed samples were masked for African ancestry.

in Argentina [27,28] is consistent with African ancestry of this origin identified in previous studies in other South America countries [33,34], and with the Eastern African ancestry identified here.

**An unrecognized genetic Native American component in Argentina.** We compared masked genotype data for admixed individuals with masked genotype data of modern Native American reference populations from South America [37,38] and a set of ancient DNA data from the region [35–37,54,55] (**DS6**; **Fig 5A**). The PCA (**Fig 5B**) and Admixture analyses (best model with *K = 3*) confirmed the three main South Native American ancestry components described by de la Fuente *et al.* (2018), and identified in our analyses at the global level.

Similar results were also observed when applying Admixture algorithm (best model with *K = 3*; **S13A Fig**). The proportion estimates for the three main Native American ancestries observed for the South America Native American individuals and for the ancient samples (**Fig 6**) highly correlate with estimates from unmasked data (**S14 Fig**), and are consistent with CAN, CCP and STF labels that we attributed. In Argentina, the three Native American ancestries are observed in almost all the admixed individuals studied. In Northeastern Argentina (NEA), STF ancestry is the most frequent. In Northwestern Argentina (NWA), CAN ancestry is not clearly dominating since STF is also observed in important proportions. In the South, CCP is observed in greater proportions.

Although migration events tend to reduce the genetic distance among these components, we still identified correlations between geographic coordinates and the proportions of each Native American ancestry component. Indeed, CAN is found in higher frequencies further

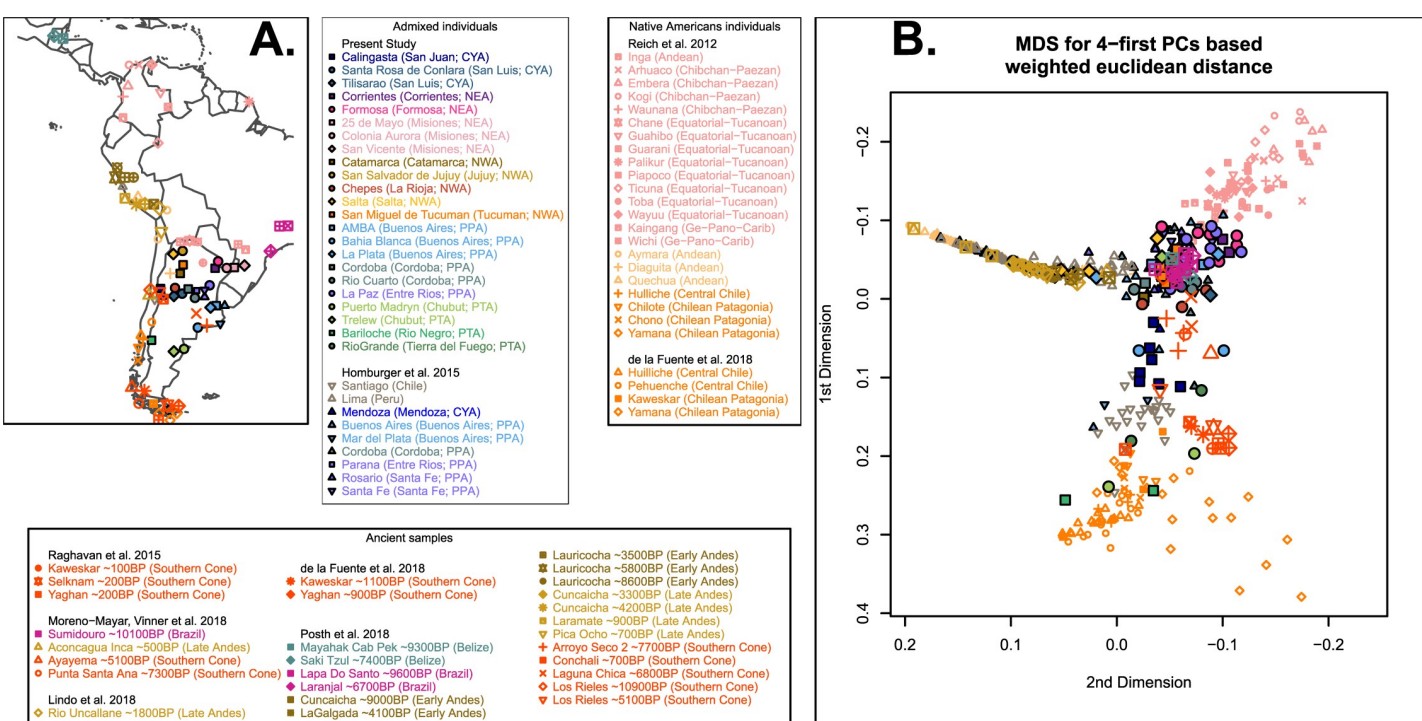

**Fig 5. Native American ancestry-specific principal component analysis.** (**A**) Localization map of the studied samples. Color and point coding matches sample groups: ancient populations are grouped following [35], modern Native American populations are grouped following [38] while Argentinean admixed sample locations are grouped according to Regions (**Fig 1**). For clarity, some geographic coordinates have been slightly changed. (**B**) Multidimensional scaling scatterplot from Euclidian distance calculated from weighted Native American ancestry specific Principal Components. Elbow method to choose the best number of PCs to compute MDS is shown in **S12 Fig**.

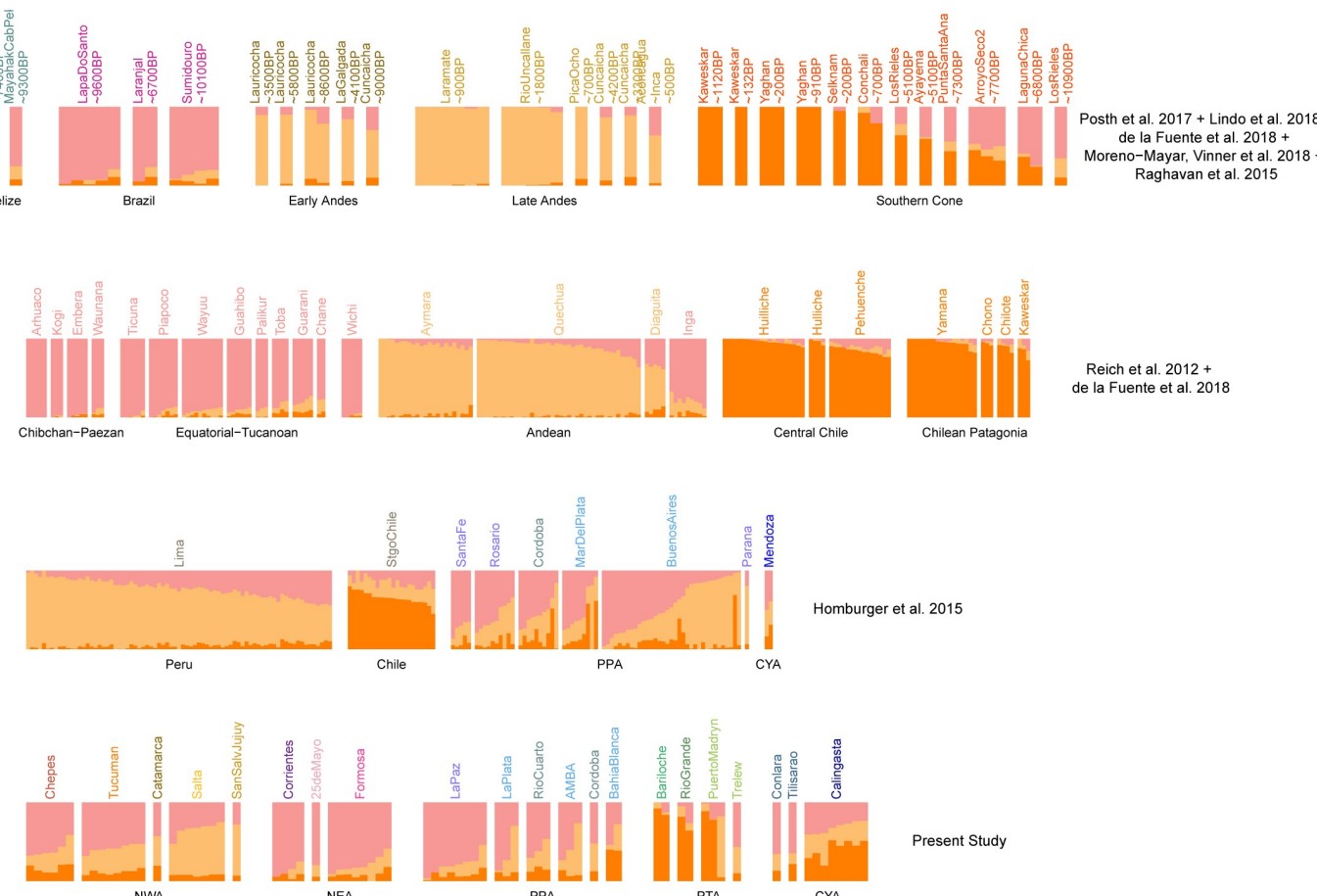

**Fig 6. Native American ancestry-specific admixture analysis.** Admixture for *K* = 3. **1st row**: Ancient samples grouped following [35]. **2nd row**: Modern Native American samples grouped following [38]. **3rd row**: Chilean, Peruvian and Argentinean admixed samples from [31]. **4th row:** Argentinean admixed samples from the present study; sample locations are grouped according to Regions (**Fig 1**); CYA: Cuyo Region; NEA: Northeastern Region, NWA: Northwestern Region; PPA: Pampean Region; PTA: Patagonia Region. Genotype data for all modern samples were masked for Native American ancestry.

North (**S15A Fig**) and West (**S15A Fig**), STF is higher further North (**S15B Fig**) and East (**S15E Fig**), while the proportion of CCP is higher further South (**S15C Fig**).

Many individuals from the Cuyo and Pampean regions of Argentina (San Juan and Córdoba provinces as well as South of Buenos Aires province) exhibit intermediate position in PCA (**Fig 5**) and mid proportion estimates with Admixture (**Fig 6**). This pattern can be interpreted as the result of a mixture between different ancestries (scenario 1) or relative limited shared history with any of them (scenario 2). In the intent to discriminate between both scenarios, we increased the number of ancestral populations to 4 and 5. Model with *K = 4* (**S13C Fig**) does not help to address this question, and model with *K* = 5 informs that scenario 2 is the most likely for Middle Holocene samples from the Southern Cone (**S13D Fig**). Models including more ancestral populations (K>5) did not allow robust ancestry proportion estimates most likely due to the masked data leveraged here. Many regions of Argentina are underrepresented in the reference panel because of the scarcity of Native American communities in most of the Argentinean territory. Altogether, PCA and admixture analyses are not sufficient to unequivocally contrast whether some Native American genetic diversity specific to Argentina is not represented in the reference panel.

In order to circumvent these limitations, we performed an objective quantitative approach based on *K*-means clustering (**S16 Fig**). We assigned the 452 modern South American individuals (the 53 ancient samples included in **DS6** were not analyzed here) to a Native American ancestry cluster. We assigned 163, 161 and 70 individuals to the clusters representing CAN, STF and CCP, respectively, and 32 individuals were assigned to a fourth cluster (**S4 Table**). The remaining 26 individuals were removed for further analyses because their group assignation was not consistent across the three clustering approaches. We acknowledge that these groups are culturally, ethnically and linguistically heterogeneous. However, we argue that analyzing such groups built from genetic similarities may provide interesting insights into evolutionary mechanisms that shaped the Native American ancestry in South America.

Most individuals from Calingasta, (located in the Northwest Monte and Thistle of the Prepuna ecoregion; San Juan Province) and from Santiago de Chile were assigned to the fourth group. The genealogical record for the Calingasta individuals attests to a local origin of their direct ancestors up to two generations ago, and they have mtDNA sub-haplogroups predominant in the Cuyo region (**S1 Table**; [56,57]). On the other hand, all the Huilliche and Pehuenche individuals from Central Chile [37] have been consistently assigned to CCP. Altogether, we decided to refer to this fourth group hereafter as Central Western Argentina hereafter (CWA).

**Relationship among the four identified Native American components.** Significant positive $f_3$ scores to test for Treeness were obtained for the six possible pairwise comparisons among the four Native American clusters identified (**Fig 7A** and **S5A Table**). In addition to confirming the differentiation among the three components described in our reference panel (CCP, CAN and STF), this analysis showed that the fourth group (CWA) most likely represents a Native American component never described in any previous study based on autosomal genetic markers. In fact, the genetic differentiation (measured with $F_{ST}$ index) between CWA and any other cluster is not lower than for other comparisons (**Fig 7B**). The lowest $F_{ST}$ was observed between STF and CAN, probably due to the fact that STF encompasses the Northern Andes region (**Fig 7B**). Moreover, the distribution of $1 -f_3(YRI; Ind1, Ind2)$ between pairs of individuals from different groups (**S17 Fig**) is an additional argument to discard a scenario of mixture (mentioned before as scenario 1).

Furthermore, $f_4$ analyses showed that (i) CAN has no particular genetic affinity with any component relative to the others; (ii) STF is closer to CAN as compared to CCP and CWA; and (iii) CWA and CCP exhibit higher genetic affinity between them than with CAN or STF (**Fig 7C**; **S5B Table**). However, a neighbor-joining analysis [58] from distances of the form $1/f_3(YRI; X, Y)$ suggests that CAN is more closely related to CCP and CWA than to STF (**Fig 7D**; **S5C Table**).

Based on these evidences, we argue that CWA may represent a Native American ancestry that diverged from CCP and established on the Eastern side of the Andes in the Cuyo region.

The existence of a specific differentiated component in the Central Western and Central regions of Argentina has been previously suggested from maternal lineages analyses accounting for the genetic relationship between these two regions [10] and the presence of specific clades underrepresented elsewhere [10,27,59]. These studies support the hypothesis of a common origin and/or important gene flow [57], and the authors referred to a meta-population with great temporal depth and differentiated from other regions in Argentina. Moreover, the ethnographic description of the populations that were settled at the moment of contact with the Spanish colonies accounts to a potential relationship between Huarpes in the Cuyo region and Comechingones in present-day Córdoba province ([60], cited by [61]). Altogether, these facts support the Central Western Argentina (**CWA**) label that we attributed to this fourth component.

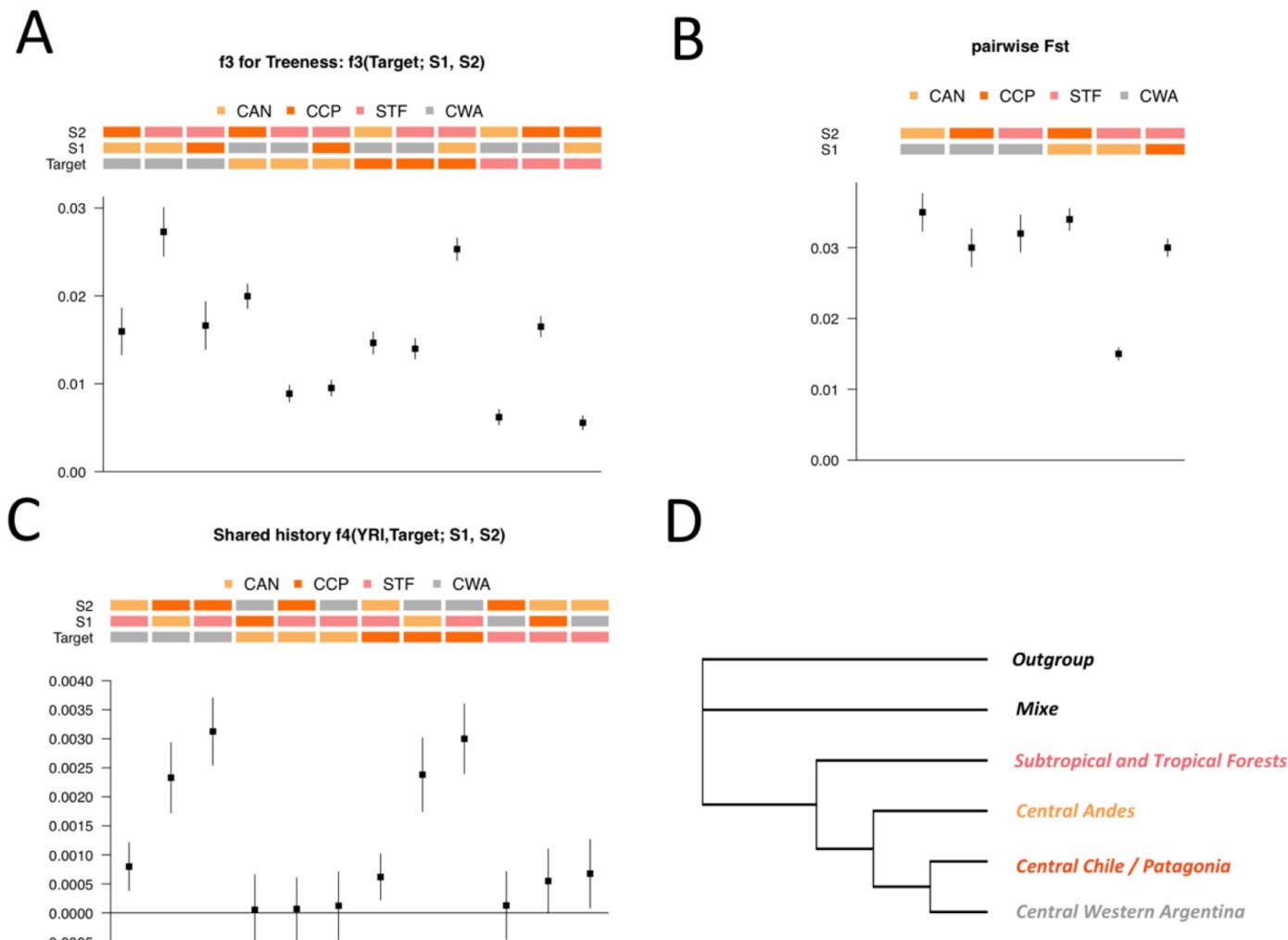

**Fig 7. Relationship among the four Native American groups identified. (A)** $f_3$(*Target; S1, S2*) to test for treeness. **(B)** $F_{ST}$ between pairs of Native American groups. **(C)** $f_4$(*YRI, Target; S1, S2*) to test whether Target shares more ancestry with S1 or S2. Since $f_4$ is symmetrical when switching *S1* and *S2*, only positive comparisons are shown. **(D)** Neighbor-joining tree estimated from distance matrix of the form $1/f_3$(YRI; X, Y). The tree was estimated using ancient sample from Upward Sun River site in Beringia (USR1) as outgroup. CAN: Central Andes; STF: Subtropical and Tropical Forests; CCP: Central Chile / Patagonia; CWA: Central Western Argentina; YRI: Yoruba from 1KGP. Vertical segments are the +/- 3 standard errors intervals in A and B.

Complementary historical facts may explain the representation of this component in Santiago de Chile and its absence in Huilliche and Pehuenche populations. First, in pre-Columbian times, it seems to have existed an ethnic boundary around the 36˚ S Latitude, since Central Andean civilizations could not expand further South [62,63]. Later, the Spanish could neither settle those territories. Second, the Spanish colonists that established in Santiago de Chile organized massive deportation of Huarpes individuals from Cuyo region to palliate the lack of indigenous workforce [64]. For example, in Santiago de Chile, in 1614, 37% of the indigenous people that lived in the suburbs were Huarpes according to the chronicler Vázquez de Espinosa [65].

The present study, in which we analyzed individuals that do not belong to any indigenous community, made possible the identification of a Native American component not previously reported from autosomal markers.

**Genetic affinity of the four Native American ancestry components with ancient populations.** We evaluated the genetic affinity of the four Native American ancestry components to ancient samples from the literature [35–37,54,55,66,67]. Graphical summaries of pairwise $f_3$-based distances are presented in S18 Fig.

When comparing the genetic affinity of a given component with the different ancient groups using either the $f_3$-*outgroup* or the $f_4$ statistics (S19F Fig and S5D and S5E Table), we identified that CAN tends to exhibit greater genetic affinity with ancient Andean populations than with other ancient groups (S19A and S19E Fig). Strikingly, the genetic affinity of this component with both Late Andes and Early Andean ancient groups would point to a genetic continuity across the whole temporal transect that the archaeological samples provide. This would suppose that the replacement of an early population arrival by a later stream of gene flow in Central Andes identified previously [35] does not fully explain the current-day genetic diversity for CAN. We observed higher genetic affinity with ancient Southern Cone groups for both CCP and CWA (S19C, S19D, S19G and S19H Fig). As for STF, we observed intermediate genetic affinity with ancient samples from both the Andes and the Southern Cone (S19B and S19F Fig). The fact that there is no ancient sample group exhibiting outstanding genetic affinity with STF points to the underrepresentation of this component in ancient samples. First, the geographical range covered by ancient samples that could represent this component is restricted to Brazil, while STF is a heterogeneous group that includes relatively isolated populations [39] from the Gran Chaco, the Amazonas and Northern Andes. Moreover, the most recent samples that could represent STF are aged ~6700BP, and gene flow with other components since then may have contributed to dissolve the genetic affinity of STF with ancient samples in Brazil analyzed here.

Then, we evaluated the relationship of the time depth of the ancient samples from either the Andes or the Southern Cone, with their genetic affinity to the modern components of Native American ancestry (Fig 8). We observed a statistically significant relationship between the age of the ancient Southern Cone samples and their genetic affinity with CCP and CWA. This means that the older the ancient sample from the Southern Cone, the lower the shared drift with CCP and CWA. On the other hand, no statistically significant relationship was identified for STF and CAN ($P = 0.523$ and $P = 0.596$, respectively; Fig 8A). These patterns could be due to a relationship between geography and the age of the ancient samples because the most recent samples are concentrated in the Southern tip of the subcontinent (Fig 5A). Moreover, the number of SNPs with genotype data tends to decrease with the age of the ancient samples due to DNA damage, and thus inducing a potential bias towards significant positive correlations. To simultaneity correct both these two putative confounding effects, we repeated the analyses but using a correction for the ancient sample age (the residuals of the linear regression between the age of the ancient samples and their geographic coordinates) and a correction for genetic affinity estimates (the residuals of the linear regression between $f_3$ and the number of SNPs to estimate it). This correction intensified the relationship described for CCP and CWA (Fig 8C). It also allowed to actually identifying significant relationships for STF and CAN. On the other hand, CAN is the only modern Native American component that exhibits a significant relationship between its genetic affinity with ancient Andean samples and their age (Fig 8B). This pattern holds after correction for geography (Fig 8D). Repeating the same analyses using $f_4$ statistics, we reached the same conclusions (S20 Fig).

**Early divergence among the four Native American components in Argentina.** Using another setting of the $f_4$ statistics (S5F Table; S21 Fig), we observed that all Southern Cone ancient samples – except Los Rieles ~10900 BP – are more similar to CCP than to CWA (S21F Fig), pointing to divergence time between those two components older than ~7700 BP. CWA is not closer to any Brazilian and Andean ancient sample as compared to CCP, reinforcing the

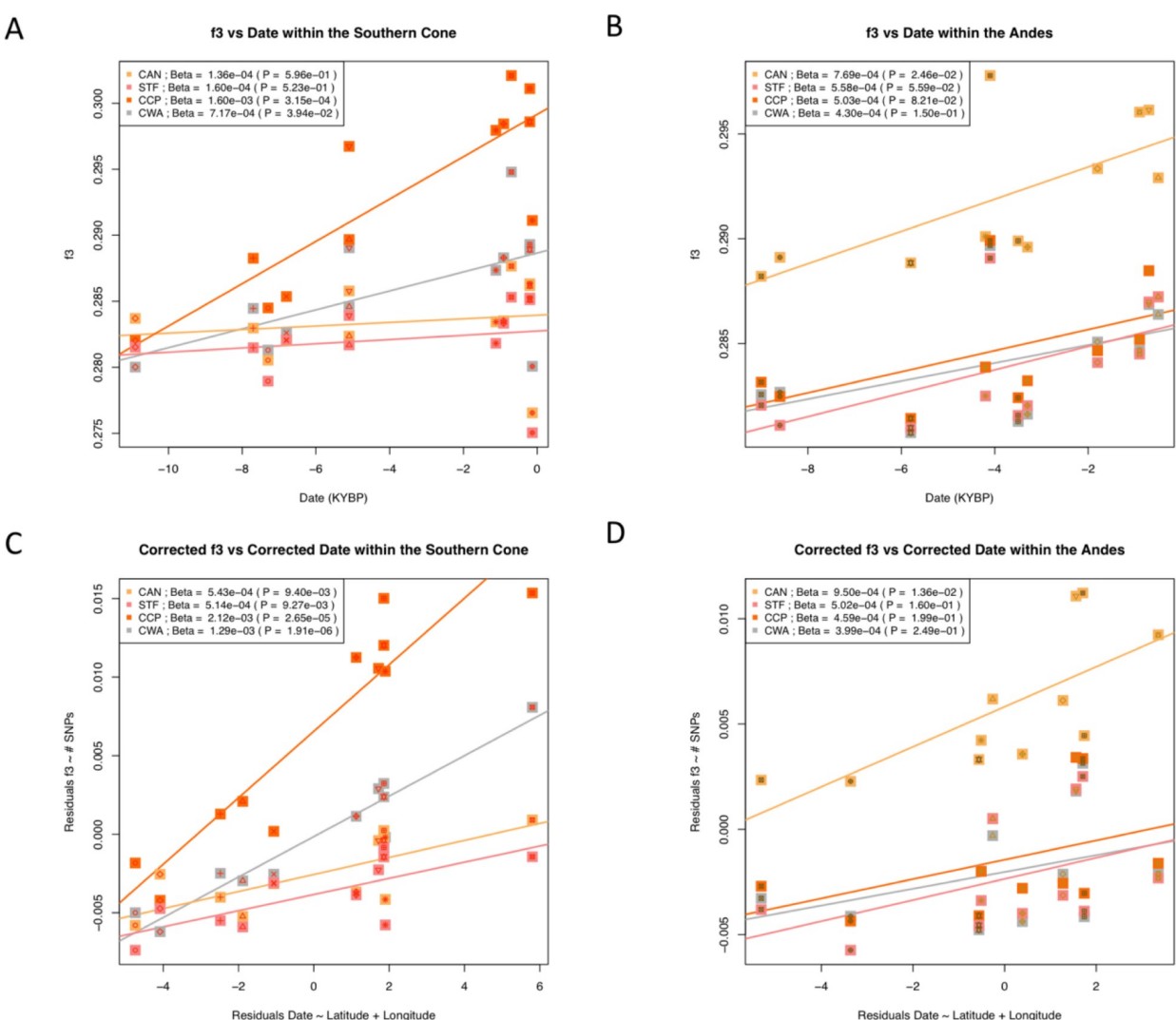

**Fig 8. Changes across time of genetic affinity of the four Native American with ancient samples.** Each point represents a $f_3$-Outgroup score of the form $f_3(YRI; X, Ancient)$ vs the age of ancient samples, where $X$ is one of the four identified modern Native American components, and *Ancient* is an ancient group. $X$ is represented by the color of the square while the symbol inside the square represents *Ancient*. The legend for plotted symbols is shown in Fig 5. **(A)** $f_3$ vs age of ancient samples from Southern Cone. **(B)** $f_3$ vs age of ancient samples from the Andes. **(C)** $f_3$ vs age of ancient samples from Southern Cone considering correction for both $f_3$ and age. **(D)** $f_3$ vs age of ancient samples from the Andes considering correction for both $f_3$ and age correction. Linear regression slopes and the associated *P*-values are shown. CAN: Central Andes; STF: Subtropical and Tropical Forests; CCP: Central Chile / Patagonia; CWA: Central Western Argentina; YRI: Yoruba from 1KGP.

idea that CWA is not a mix of CCP, CAN and STF related ancestries. Moreover, we observed that all the ancient groups of the Southern Cone have greater genetic affinity with CCP than with CAN and STF (**S21B and S21D Fig**). Ancient groups of the Southern Cone (with the exception of Los Rieles ~10900 BP) tend to exhibit higher genetic affinity with CWA than with STF (**S21C Fig**). In addition, the comparisons including STF and CWA also point to a closer genetic relation between ancient groups from the Southern Cone with CWA, although statistical significance is reached only for Late Holocene Patagonia samples.

All the ancient samples from the Andes, independently of their previous assignment to the Early or Late population stream [35], are genetically more similar to CAN than to any of the three other components (**S21A, S21D and S21E Fig**). These results support further the

hypothesis that the ancestry associated with an early Andean population arrival remains present to some extent in modern Central Andean populations and has not been totally replaced by a later stream of gene flow. Moreover, the archaeological samples in Brazil have higher genetic affinity with CAN than with CWA and CCP (not statistically significant for the latter comparisons). This suggests that divergence of STF with CCP and CWA could have occurred before the divergence of STF with CAN (also observed in **S18A Fig**).

Altogether, the trends depicted by these analyzes are expected under a model of early divergence among the four components. Our results also support the hypothesis of genetic continuity over long period of time for the different components after their split.

In order to get insights into the past genetic influence among the four components since their divergence, we applied a last $f_4$-statistics analysis (**S5G Table**; **S22 Fig**). We observed that the influence of CAN appears to have been more pronounced into STF component than into CWA and CCP components (**S22B, S22G and S22I Fig**). The genetic affinity of CAN with STF, CCP and CWA is reduced as compared to both ancient Early and Late Andes samples (**S22A**, **S22H and S22J Fig**), implying that that the genetic legacy of Andean ancient populations into CAN remains important nowadays. The mutual influence of CCP and CWA components seems to have been important in remote times (**S22K and S22L Fig**). However, the available ancient samples provide much more precise information on the influence of CCP into CWA component than the other way around. Indeed, CWA does not exhibit higher genetic affinity to any ancient group relative to CCP (**S22K Fig**). The archaeological record for which genetic data has been generated misrepresents CWA since its early divergence with CCP, as well as the common ancestors specific to these two components.

## Conclusions

European ancestry in modern Argentinean populations is mostly explained by Southern European origins, although we identified several individuals with Northern European ancestry. African ancestry in Argentina traces its origins from three main components. Although the Bantu-influenced and Western components are clearly the most represented, we also found that an Eastern origin explains some of the African ancestry in Argentina and represents up to ~30% of the African ancestry for some Argentinean individuals. Our work shows that studying more admixed individuals, with a particular focus on extending the geographical coverage of the Argentinean territory, would help to identify the genetic legacy from secondary migration streams, and thus to get a better representation of the complex origins of African and European ancestries in the country.

Concerning Native American ancestry, we concluded that Argentinian populations share, in varying proportions, from three distinct components previously described: Central Andes, Central Chile/Patagonia and Subtropical and Tropical Forests. Moreover, we present evidence supporting the hypothesis that the Native American ancestry in the Central Western region of Argentina may derive from a fourth component that diverged early from the other Native American components, and maintained a tight link with the Central Chile/Patagonia component. This relation is not explained by a putative contribution of admixed individuals from Santiago de Chile (**S23 Fig**). Having identified this component from admixed individuals demonstrates that focusing only on indigenous communities is insufficient, at least in Argentina, to fully characterize the Native American genetic diversity and decipher the pre-Columbian history of Native Americans. Indeed, most indigenous communities have been culturally annihilated and invisibilized [68,69] to the point that several Argentinean regions were considered "Indian free" in the mid-20[st] century [70]. However, the cultural incorporation did not necessarily imply a biological extinction. Although studies based on samples from indigenous

communities [37–39] provide decisive information to understand the evolutionary history of Native American ancestry, alternative strategies must be considered to fill this gap in the effort to more fully describe the Native American ancestry (e.g. see [71]). Studying admixed individuals can be complex, and leveraging a pure statistical approach, we grouped individuals from rather culturally, ethnically, linguistically, and genetically heterogeneous groups to represent the four Native American components discussed here. Yet, the present study provides useful insights into the routes followed by the main population arrivals in the Southern Cone (**S24 Fig**).

Further efforts are needed to better characterize the Native American ancestry component identified in the Central Western region of Argentina. Particularly, we encourage future studies to confirm the tentative geographical label that we suggest here, and to estimate its influence in the region. Besides these specific questions, many other general questions remain to be answered to better understand the pre-Columbian population dynamics in the Southern Cone such as the time and place of the splits among the components described here, and the extent of genetic exchanges among them. More genotype data for ancient samples, modern indigenous communities and admixed individuals, particularly in Central, Northeastern and Patagonia regions of Argentina, would help to decipher these issues.

The genomic characterization of populations is an unavoidable practice for many issues ranging from the understanding of our biological heritage, the rational use of biobanks, the definition of an adequate reference genome, the estimation of polygenic risk scores, the study and treatment of simple and complex diseases, and the design of a national program of genomic medicine in our country. This study is a joint effort of Argentinean institutions funded by the national scientific system, and represents the first milestone of the *Consorcio Poblar*, a national consortium for creating a public reference biobank to support biomedical genomic research in Argentina [72]. Genomic knowledge of local populations should be a priority for developing countries to achieve an unbiased representation of diversity in public databases and the scientific development in periphery countries.

## Material and methods

### Studied samples

We genotyped 94 individuals with the Axiom LAT1 array (Affymetrix) from 24 localities and 17 provinces across Argentina (**Fig 1**). These samples were selected among 240 collected by different population genetics groups (Consorcio PoblAR) during past sampling campaigns with a biological anthropology focus. According to the available information (e.g. interviews, genealogical information, etc.), each PoblAR research group selected for this study some samples, maximizing the odds that they come from individuals with greater Native American ancestry. For example, surnames were used as a proxy to achieve this objective and the permanence of ancestors in national territory has been another variable that was taken into account. The analyzed 94 samples were also selected to ensure an extended geographical range and were included when they presented sufficient DNA concentration and Native American maternal lineage. Moreover, among the males, we prioritized those with Native American paternal lineage.

All saliva and blood samples were collected under written informed consent to participate to this study. The informed consents for research use were approved by several Ethics Committees from (i) Hospital Zonal de Puerto Madryn (Resolución 009/2015), (ii) Hospital Zonal de San Carlos de Bariloche (Resolución 1510/2015), (iii) Hospital Provincial de Pediatría Dr. Fernando Barreyro, (iv) Investigaciones Biomédicas IMBICE (RENIS CE000023), (v) Provincia de Jujuy, (vi) Hospital Italiano of Buenos Aires (protócolo 1356/09); and (vii). Centro de

Educación Médica e Investigaciones Clínicas "Norberto Quirno" (Resolución 612/2018). The biological samples were coded and anonymized, as per the Argentina National Law 25.326 of Protection of Personal Data.

87 samples and 791,543 autosomal variants passed the standard Affymetrix genotyping Quality Controls (**S1 Table**).

Most of the genotype data processing was performed using in-house scripts in R [73] and perl [74], leveraging *plink2* [75], *vcftools v1.13* [76], and *bedTools v2* [77].

We compiled the genotype data for the 87 Argentinean samples with different genotype data available in the literature. We focused our study on biallelic SNPs (removing indels and SNPs with more than 2 alleles). Any putative inconsistent strand had been fixed processing to the relevant flip, filtering out any SNP with ambiguous genotype (A/T, G/C).

Cryptic relatedness among samples were assessed using King software [78]. To avoid any 1$^{st}$ degree relationship, we filtered out individuals, minimizing the total number of removals. No admixed individuals had been removed at this step.

We thus build different dataset arrangements (named **DS<n>**) that we analyzed through this work (**S2 Table**).

## Argentinean genetic diversity within a worldwide context

To analyze genetic diversity in Argentina within a worldwide context, we built the Dataset1 (**DS1**). This dataset contains 87 Argentinean samples, 654 African, 503 European and 179 South American samples from 1KGP [42], 54 modern unrelated Chilean samples [37], and 161 Native-American individuals from South America [38]. Moreover, **DS1** included genotype data from [31] which consists in 82 individuals from Lima (Peru), 27 from Santiago de Chile, and 161 from Argentinean urban centers.

We filtered out any variant and individual exhibiting in the compiled data set more than 2% and 5% of missing genotypes (—geno 0.02 and—mind 0.05 flags in plink 1.9), respectively, as well as Minor Allele Frequency below 1% (—maf 0.01 flag in plink 1.9). The filtered data was then pruned for Linkage-Disequilibrium (—indep-pairwise 50 5 0.5 flag in plink 1.9). The combined data set has a total intersection of 59,237 SNPs and 1,908 individuals. With this curated data we performed Principal Component Analyses [41] (**S1 Fig**) and Admixture [43] (**Fig 2**, **S2 Fig and S3 Table**).

## Local ancestry

Local ancestry analyses rely on haplotype reconstruction (phasing) and require high SNP density. Since, admixed individuals of interest were genotyped on different microarray platforms, we decided to perform two separate local ancestry analyses on two different data sets (**DS2** and **DS3**). **DS2** consists of 87 Argentinean individuals from the present study, and 54 Chilean individuals [37], all of them genotyped with the Axiom LAT1 microarray. **DS3** includes 175 Argentinean, 27 Chilean, and 119 Peruvian genotyped with the Illumina OMNI1 microarray [31]. Both **DS2**, **DS3** were merged with 1KGP data consisting in 503 phased reference samples for each of the African and European genetic ancestry, and 347 Latin American individuals.

The 87 Argentinean, and the 54 Chilean samples were phased with shapeIT2 [79,80]. The genetic map, and 5,008 haplotypes panel provided by 1KGP were downloaded from http://mathgen.stats.ox.ac.uk/impute/1000GP_Phase3/. Algorithm, and model parameters were used by default, filtering out monomorphic SNPs, and those with more than 2% of missing genotype. We obtained phased genotype data for 608,501 autosomal SNPs. This data was then merged with phased 1KGP genotype data for African, American and European samples described before. Since this data set derived from DS2, we call it **DS2p** (phased DS2).

The 175 Argentinean, 27 Chilean and 119 Peruvian samples from [31] were phased separately using the same procedure with shapeIT2. After filtering for missing genotypes and merging with phased 1KGP genotype data for African, American and European samples we obtained phased data for 694,626 autosomal SNPs. Since this data set derived from DS3, we call it **DS3p** (phased DS3).

In **DS2** and **DS3** we ran Admixture using K parameter minimizing the Cross-Validation score. We used individuals with more than 99% of Native American ancestry as references for local ancestry estimation. For **DS2** we used $K = 7$, and Native American ancestry was defined as the sum of the two American specific components observed (**S25 Fig**). According to this criterion, 48 individuals were assigned as Native American reference. All the other American samples were defined as Admixed.

For **DS3**, we used $K = 5$, and Native American ancestry was defined as the single American specific component observed (**S26 Fig**). According to this criterion, 19 individuals were assigned as Native American reference. All the other American samples were defined as Admixed.

RFMix v2 ([44] downloaded at https://github.com/slowkoni/rfmix on 15th August 2018) was run on **DS2P** and **DS3P** separately using parameter settings similar to [81]. The reference panels consist of the African and European samples from 1KGP, as well as Native American individuals identified through Admixture procedure described before. We used 1 Expectation-Maximization iteration (-e 1), actualizing the reference panel in this process (—reanalyze-reference). We used CRF spacing size and random forest window size of 0.2 cM (-c 0.2 and -r 0.2). We use a node size of 5 (-n 5). We set the number of generations since admixture to 11 (-G 11) considering the estimates from [31]. The forward-backward output was then interpreted to assign allele ancestry to the one exhibiting major posterior probability, conditioning that it was greater than 0.9. Otherwise, the allele ancestry was assigned to Unknown (UKN). As a sanity check, the global ancestry proportions estimated through this RFMix analysis were compared with those obtained with Admixture software. The global ancestry proportion estimates obtained by both procedures matched very well: spearman's correlation greater than 0.9 in American samples for any of the 3 continental ancestries (**S27 Fig**). Moreover, the ancestry ditypes assigned in admixed individuals from 1KGP (included in both DS2p and DS3p) were highly consistent between both independent masking procedures (**S28 Fig**).

## Ancestry specific population structure

In order to analyze the ancestry-specific population structure we masked the data, i.e. for each individual, we assigned missing genotype for any position for which at least one of the two alleles was not assigned to the relevant ancestry. In other words, to study ancestry A, we kept for each individual, regions exhibiting ancestry A on both haplotypes (ditypes) as illustrated in **S5 Fig**

**European ancestry specific population structure.** To study European ancestry specific population structure, we analyzed together masked data for this ancestry for Colombian individuals from 1KG and individuals from **DS2P** and **DS3P** excluding individuals from Chilean Native American communities [37]. This data was merged with a set of reference individuals with European ancestry [46], which is a subset of the POPRES dataset [45]. We call this data set as **DS4**. We removed individuals with less than 30% SNPs with the ancestry ditypes (—mind 0.7 with *plink* 1.9). We also removed SNPs with more than 50% of missing genotypes (—geno 0.5 with *plink* 1.9). Thus, **DS4** contains 132 modern Argentinean individuals (29 from the present study and 103 from [31]), 17 individuals from Santiago de Chile, 4 from Lima and 74 from Colombia [42]. **DS4** encompasses 29,347 SNPs of which 27,634 remained after LD-pruning (—indep-pairwise 50 5 0.5 flag in *plink2*).

*Smartpca* from Eigensoft v7.2.0 was run on **DS4** [41] with the *lsqproject* option ON. We report the PCA results summarized into a 2-dimensional space by applying Multidimensional Scaling on weighted Euclidian distance matrix for the first N PCs. We weighted each PC by the proportion of variance it explains. We selected the N most informative PCs according to the Elbow method on the proportion of explained variance. Admixture [43] was run with *K* ranging from 2 to 10 with cross-validation procedure.

**African ancestry specific population structure.** To study African ancestry specific population structure, we analyzed together masked data for this ancestry for individuals from **DS2P** and **DS3P**. This data was merged with a compilation of reference individuals with African ancestry from [42,48–50]. We removed African individuals with less than 99% of African ancestry when comparing them to the 2504 individuals from 1KGP (Admixture with *K* = 7 minimizing cross-validation score). We thus reduced the African reference to 1685 individuals. We call as **DS5** the data set containing both the masked data for admixed South American individuals and African reference individuals.

We removed SNPs with more than 10% of missing genotypes (—geno 0.1 with *plink* 1.9), and individuals with less than 5% of the ancestry ditypes (—mind 0.95 with *plink* 1.9). Thus, **DS5** contains, 26 modern Argentinean individuals (all from the present study), and 12 individuals from Lima (*9*). **DS5** consisted in 137,136 SNPs, of which 128,086 remained after LD-pruning (—indep-pairwise 50 5 0.5 flag in *plink2*).

PCA and Admixture were performed as for European ancestry specific population structure analyses (described before).

**Native American ancestry specific population structure.** To study Native American ancestry specific population structure, we analyzed together masked data for this ancestry for individuals from **DS2P** and **DS3P**. This data was merged with pseudo-haploid data for ancient samples within South and Central America [35–37,54,55], as well as with masked data for Native American individuals from [38]. The pseudo-haploid data for ancient samples was downloaded from the Reich Laboratory webpage (https://reich.hms.harvard.edu/downloadable-genotypes-present-day-and-ancient-dna-data-compiled-published-papers) the 15th of May 2019. For each sample, we used annotations (geographic coordinates and approximate date) from the metafile provided at the same url.

We call this data set **DS6.** We removed individuals with less than 30% SNPs with the ancestry ditypes (—mind 0.7 with *plink* 1.9). We also removed SNPs with less than 50% individuals with the ancestry ditypes (—geno 0.5 with *plink* 1.9). **DS6** contains 146 modern Argentinean individuals (74 from the present study and 72 from [31]), 22 individuals from Santiago and 77 from Lima, along with 207 Native South American individuals from [37,38] and 53 ancient samples. **DS6** encompasses 47,003 SNPs, of which 39,423 remained after LD-pruning (—indep-pairwise 50 5 0.5 flag in *plink2*).

PCA and Admixture were performed as for European ancestry specific population structure analyses (described before), with the difference that the *poplistname* option was set for *smartpca* in order to estimate the PC using only modern samples and project the ancient samples.

**Statistical assignation of modern South American individuals to Native American components.** Given a distance matrix among modern individuals from **DS6**, we performed *K-means* clustering with the number of clusters ranging from 2 to 20 and selected the *K-means* output minimizing the Bayesian Information Criterion (BIC). This procedure was applied to three different distance matrix: (i) Weighted Euclidean distance based on the two first PCs from Principal Component Analysis (ii) Euclidean distance based on the ancestry proportions estimated from Admixture (*K* = 3) and (iii) $1 - f_3(Ind1, Ind2, Yoruba)$, where *Ind1* and *Ind2* are two individuals. In the three cases, the *K*-means procedure minimized the BIC when

considering 4 clusters. These clusters correspond to the three Native American Component discussed in this paper (CAN, CCP and STF), along with one laying in-between, which we finally attributed to Central Western Argentina. However, individual assignment to one of these four clusters was not totally consistent according to the distance matrix we used. To obtain a robust assignation, an individual was assigned to a given cluster when it consistently belonged to it across the three $K$-means procedures, otherwise it had been removed for following analyses (**S16 Fig**). The cluster assignation for each individual is given in **S4 Table**.

We computed pairwise $F_{ST}$ among the groups using smartpca [41] with fstonly and lsqproject set to YES, and all the other settings left at default. Standard errors were estimated with the block jackknife procedure [41].

**$F$-statistics to infer relationship between the four Native American components in Argentina and their genetic affinity with ancient populations.** Starting from genotype data for the individuals in **DS6**, we also included genotype data from Yoruba (YRI) population from [42], Mixe population from [38] and, pseudo-haploid data for Anzick individual from the Clovis culture [67] and USR1 individual from Upward Sun River in Beringia [66]. The resulting data set is called **DS7**. Within **DS7,** we grouped modern individuals according to their assigned Native American group and we removed those with inconsistent assignation (**S16 Fig** and **S4 Table**). Thus, **DS7** contains 426 modern individuals, 55 ancient samples, 108 Yoruba and 17 Mixe individuals. **DS7** encompasses 88,564 SNPs. Note that we did not apply any SNP filtering overall **DS7** in order to maximize the number of SNPs included in each group comparison considered.

Using modern individuals from **DS7**, and considering any possible combination of the four groups identified (STF, CCP, CAN and CWA), we computed the $f_3$ statistics [82] in the form of $f_3(Target; S1, S2)$. This allowed to contrast whether $Target$, $S1$ and $S2$ could be organized in the form of a phylogenetic tree $(positive\ f_3)$ or whether the $Target$ group is the result of mixture between $S1$ and $S2$ groups (negative $f_3$). We also computed $f_4(YRI, Target; S1, S2)$ to test whether group $Target$ shares more evolutionary history with group $S1$ (negative $f_4$) or group $S2$ (positive $f_4$). Moreover, we reconstructed the Neighbour-joining tree from the matrix of distances with Phylip v3.2 [58] and $USR1$ as outgroup. The distances were estimated as $1/f_3$-$outgroup(YRI; X, Y)$.

We computed $f_3$-Outgroup and $f_4$ statistics in order to estimate the genetic affinity of the four Native American components with ancient populations from South and Central America. We computed the $f_3$-$outgroup$ and $f_4$ statistics in the form of $f_3(YRI; X, Ancient)$ and $f_4(YRI, X; USR1, Ancient)$, where $X$ the represent the cluster containing all individuals assigned to a given Native American ancestry component.

We also computed $f_4$ of the form $f_4(YRI, Ancient; X, Y)$. This $f_4$ setting allowed testing which of $X$ or $Y$, each referring to one of the four Native American components. shares more ancestry with a given $Ancient$ group. $X$ and $Y$.

We finally computed $f_4$ of the form $f_4(Ancient, X; Y, YRI)$ to test whether a given modern Native American component $Y$ shares exhibit closer genetic affinity with a given $Ancient$ group (negative $f_4$) or with another modern Native American component $X$ (positive $f_4$).

All the results based on $F$-statistics are listed in **S5 Table**. We assessed significance of a comparison considering 3 standard errors ($|Z| > 3$) from block jackknife procedure [82].

## Geographical maps

All maps have been generated in R using, *maps* package, (https://cran.r-project.org/web/packages/maps/index.html). **S24 Fig** includes a raster downloaded at https://www.naturalearthdata.com/ and integrated in R with *raster* (https://CRAN.R-project.org/package=raster) and *RStoolbox* (http://bleutner.github.io/RStoolbox) packages.

## Supporting information

**S1 Fig. Principal component analysis in a worldwide Context. A:** PC2 vs PC1; **B:** PC4 vs PC3; C: PC6 vs PC5. The percentage of variance explained by each principal component (PC) is given. Each point represents an individual following the color and point codes given in legend.
(TIF)

**S2 Fig. Admixture analysis in a worldwide context. (A)** Cross-Validation score for Admixture runs on the worldwide meta dataset (**DS1**) with $K$ from 3 to 12. **(B-G)** Admixture results with $K = 3$ to $K = 8$. 1KGP: 1000 Genomes Project; CYA: Cuyo Region; NEA: Northeastern Region, NWA: Northwestern Region; PPA: Pampean Region; PTA: Patagonia Region.
(PDF)

**S3 Fig. Comparison of different admixture continental ancestry proportion estimates in a worldwide context.** Comparison of the African, European and Native American ancestry proportion estimates obtained with Admixture models with $K = 3$ and $K = 8$ applied to **DS1**. **(A)** African ancestry proportions for K = 3 are as observed in green in **S2B Fig**, while for $K = 8$ they are estimated as the sum of the three greenish colors observed in **Main Fig 2**. **(B)** European ancestry proportions for $K = 3$ are as observed in blue in **S2B Fig**, while for $K = 8$ they are estimated as the sum of the two bluish colors observed in **Main Fig 2**. **(C)** Native American ancestry proportions for $K = 3$ are as observed in orange in **S2B Fig**, while for $K = 8$ they are estimated as the sum of the three reddish colors observed in **Main Fig 2**.
(PDF)

**S4 Fig. Correlation between eigenvectors and ancestry proportion estimates from analyses in a worldwide context.** Comparison of ancestry proportion estimates from Admixture model with $K = 8$ and the 6 first Principal Components (PCs) in **DS1**. **(A)** PC1 vs African ancestry proportions (estimated as the sum of the three greenish colors observed in **Main Main Fig 2**). **(B)** PC2 vs European ancestry proportions (estimated as the sum of the two bluish colors observed in **Main Fig 2**). **(C)** PC2 vs Native American ancestry proportions (estimated as the sum of the three redish colors observed in **Main Fig 2**). **(D)** PC6 vs Bantu-influenced ancestry proportions (dark olive green in **Main Fig 2**). **(E)** PC6 vs Western African ancestry proportions (dark green in **Main Fig 2**). **(F)** PC4 vs Southern European ancestry proportions (light blue in **Main Fig 2**). **(G)** PC4 vs Northern European ancestry proportions (dark blue in **Main Fig 2**). **(H)** PC3 vs Cenral Chile / Patagonia ancestry proportions (orange in **Main Fig 2**). **(I)** PC3 vs Central Andes ancestry proportions (yellow in **Main Fig 2**). **(J)** PC5 vs Subtropical and Tropical Forests ancestry proportions (pink in **Main Fig 2**).
(PDF)

**S5 Fig. Example of a local ancestry output. (A)** RFMIX output for a given admixed individual. **(B)** Masked genotype showing ditypes of Native American (red), European (blue) and African (green) ancestry. Gaps are represented in grey and regions with unassigned ancestry (Unknown) are in black.
(TIFF)

**S6 Fig. Choice of the number of principal components from European ancestry-specific principal component analysis.** Elbow method to determine which PC minimizes the angle of the curve from the chart "Percentage of variance explained *versus* Number of PCs"
(PDF)

**S7 Fig. European ancestry specific admixture analysis.** (**A**) Cross-Validation scores for *K* from 2 to 10. (**B**) Admixture for *K* = 2. (**C**) Admixture for *K* = 3. CYA: Cuyo Region; NEA: Northeastern Region, NWA: Northwestern Region; PPA: Pampean Region; PTA: Patagonia Region.
(PDF)

**S8 Fig. Comparision of different European ancestry proportions in South America.** Comparison of ancestry proportion estimates from European Ancestry Specific Admixture Analyses (*K* = 3) among samples from the Argentina, Chile and Colombia (**A**) Southeaster/Italian ancestry (light blue in **S7C Fig**). (**B**) Iberian ancestry (turquoise in **S7C Fig**). (**C**) Northern Ancestry (dark blue in **S7C Fig**). *P*-value of the Wilcoxon test for each pairwise comparison is shown.
(PDF)

**S9 Fig. African ancestry-specific principal component analysis.** (**A**) Localization map of the 1685 reference samples with >99% of African ancestry. (**B-C**) Principal Components performed using the African reference samples (represented as in panel **A**), and South American samples masked for African ancestry.
(TIF)

**S10 Fig. African ancestry-specific admixture analysis.** (**A**) Cross-validation scores *K* from 2 to 10. (**B**) Admixture for *K* = 2. (**C**) Admixture for *K* = 3. (**D**) Admixture plots for *K* = 4. CYA: Cuyo Region; NEA: Northeastern Region, NWA: Northwestern Region; PPA: Pampean Region; PTA: Patagonia Region.
(PDF)

**S11 Fig. Correlation between eigenvectors and ancestry proportion estimates from African ancestry specific analyses.** Comparison of ancestry proportion estimates from Admixture model with *K* = 5 and some Principal Components (PCs) in admixed samples from **DS5.** (**A**) PC3 vs Western African ancestry proportions (yellow in **S10 Fig**). (**B**) PC3 vs Bantu-influenced ancestry proportions (blue in **S10 Fig**). (**C**) PC4 vs Eastern African ancestry proportions (orange in **S10 Fig**).
(PDF)

**S12 Fig. Choice of the number of principal components from Native American ancestry-specific principal component analysis.** Elbow method to determine which PC minimizes the angle of the curve from the chart "Percentage of variance explained *versus* Number of PCs"
(PDF)

**S13 Fig. Native American ancestry-specific admixture analysis.** (**A**) Cross-validation scores *K* from 2 to 10. (**B**) Admixture for *K* = 2. (**C**) Admixture for *K* = 4. (**D**) Admixture plots for *K* = 5. CYA: Cuyo Region; NEA: Northeastern Region, NWA: Northwestern Region; PPA: Pampean Region; PTA: Patagonia Region.
(PDF)

**S14 Fig. Comparison of Native American ancestry proportion estimates obtained with admixture on unmasked data (*K* = 8) and masked data (*K* = 3).** Unmasked and masked data refers to **DS1** and **DS6**, respectively. Spearman correlation coefficients and associated *P*-values are shown. CAN: Central Andes; STF: Subtropical and Tropical Forests; CCP: Central Chile / Patagonia.
(PDF)

**S15 Fig. Correlation of Native American ancestry proportions and geographic coordinates in Argentina.** (**A**) Central Andes ancestry proportions vs Latitude. (**B**) Central Andes ancestry proportions vs Longitude. (**C**) Subtropical and Tropical Forests ancestry proportions vs Latitude. (**D**) Subtropical and Tropical Forests ancestry proportions vs Longitude. (**E**) Central Chile/Patagonia ancestry proportions vs Latitude. (**F**) Central Chile/Patagonia ancestry proportions vs Longitude. Linear regression slopes and the associated *P*-values are shown. (PDF)

**S16 Fig. Individual assignation to a Native American ancestry cluster.** Consensus cluster assignation of South American individuals based on three *K-means* procedures run with different pairwise distances among individuals. (**Top**) *K-means* results using Ancestry-Specific PCA and Admixture (ASPCA and AS-Admixture), and $f_3$ results to compute pairwise distances. Individuals are represented as in **Main Fig 5**. Insets: BIC score for number of clusters set to *K-means* ranging from 2 to 20. In all the three cases, *K*-means BIC was minimized when considering 4 clusters. (**Bottom**) Same as top with point colors corresponding to the assigned cluster. (TIF)

**S17 Fig. Pairwise genetic affinity among individuals assigned to different groups.** Boxplots for *1-$f_3$(YRI; Ind1, Ind2)*, where *Ind1* and *Ind2* are two individuals belonging to Group 1 and Group 2, respectively. The groups are either the fourth Native American components identified or ancient Middle Holocene Southern Cone groups. For clarity, boxplot outliers are not shown. YRI: Yoruba from 1KGP. (PDF)

**S18 Fig. Graphical visualization of pairwise genetic distances among modern and ancient groups in South America.** (**A**) Neighbor-joining tree from distances of the form $1/f_3$*(YRI; X, Y)*. USR1 from Ancient Beringia was used as *outgroup* (**B**) Multidimensional-scaling from distances of the form $1-f_3$*(YRI; X, Y)*. Each group is represented as appearing in the leaf of (A). USR1 and Anzick-1 were not considered in (B). YRI: Yoruba from 1KGP. (TIF)

**S19 Fig. Genetic affinity of the four Native American components with ancient groups.** (**A-D**) $f_3$*(YRI; X; Ancient)*. (**E-H**) $f_4$*(YRI, X; Ancient Beringia, Ancient)*. (**A**) and (**E**): with Central Andes (CAN) as *X*. (**B**) and (**F**): With Subtropical and Tropical Forests (STF) as *X*. (**C**) and (**G**): With Central Chile / Patagonia (CCP) as *X*. (**D**) and (**H**): With Central Western Argentina (CWA) as *X*. YRI: Yoruba from 1KGP; *Ancient Beringia*: USR1 individual from [66]; *X*: Native American component in Argentina (one plot per *X)*. *Ancient*: ancient group labeled on the *x*-axis and represented with a point/color scheme as in **Main Fig 5**. Vertical segments are the +/- 3 standard errors intervals. (PDF)

**S20 Fig. Changes across time of genetic affinity of the four Native American components with ancient groups.** Each point represents a $f_4$ score of the form $f_4$*(YRI, X; Ancient Beringia, Ancient)* vs the age of ancient sample, where *X* is one of the four identified Native American components, and *Ancient* is an ancient group. *X* is represented by the color of the square while *Ancient* is represented by the point within the square. The point code of the ancient samples is shown in **Main Fig 5**. *Ancient Beringia*: USR1 individual from [66]. (**A**) $f_4$ vs age of ancient samples from Southern Cone. (**B**) $f_4$ vs age of ancient samples from Andes. (**C**) $f_4$ vs age of ancient samples from Southern Cone considering correction for both $f_4$ *and* age. (**D**) $f_4$ vs age of ancient samples from Andes considering correction for both $f_4$ and age. Linear regression slopes and the associated *P*-values are shown. CAN: Central Andes; STF: Subtropical and

Tropical Forests; CCP: Central Chile / Patagonia; CWA: Central Western Argentina; YRI: Yoruba from 1KGP.
(TIF)

**S21 Fig. Comparison of genetic affinity of an ancient group to a Native American component relative to another.** $f_4$ *(YRI, Ancient, X, Y)* where *X* and *Y* are two of the four identified Native American components (one plot per *X-Y* combination), and *Ancient* is ancient group labeled on the *x*-axis and represented with a point/color scheme as in **Main Fig 5**. Vertical segments are the +/- 3 standard errors intervals. Note this setting for $f_4$ statistics is symmetrical when switching *X* and *Y*.
(PDF)

**S22 Fig. Comparison of genetic affinity of a Native American component to another relative to an ancient group.** $f_4$*(Ancient, X; Y, YRI)* where *X* and *Y* are two of the four identified Native American components (one plot per *X-Y* combination), and *Ancient* is ancient group labeled on the *x*-axis and represented with a point/color scheme as in **Main Fig 5**. Vertical segments are the +/- 3 standard errors intervals. CAN: Central Andes; STF: Subtropical and Tropical Forests; CCP: Central Chile / Patagonia; CWA: Central Western Argentina; YRI: Yoruba from 1KGP.
(PDF)

**S23 Fig. Removing admixed Santiago de Chile individuals to compute *F*-statistics does not affect the results.** Admixed individuals from Santiago de Chile were removed to perform the analyses presented in this figure. **(A)** $f_3$*(Target; S1, S2)* to test for treeness; **(B)** $f_4$*(YRI, Target; S1, S2)* to test whether Target shares more ancestry with S1 or S2; **(C)** $f_3$*(YRI; CWA; Ancient)*; **(D)** $f_4$*(YRI, CWA; Ancient Beringia, Ancient)*; **(E)** $f_4$*(Ancient, CCP; CWA, YRI)*; **(F)** $f_4$*(Ancient, CWA; CCP YRI)*; **(G)** $f_4$*(YRI, Ancient; CWA, CCP)*. CCP: Central Chile / Patagonia.
(PDF)

**S24 Fig. Schematic routes for the main population arrivals in the Southern Cone.** Each arrow represents one of the four components discussed throughout the article. Neither the time and place of the splits among these components nor gene flow among them have been addressed in this study.
(PDF)

**S25 Fig. Admixture analyses in DS2 to define European, African and Native American reference individuals for local ancestry analyses.** **(A)** Cross-validation scores from *K* = 3 to *K* = 10. **(B)** Admixture for *K = 7*.
(PDF)

**S26 Fig. Admixture analyses in DS3 to define European, African and Native American reference individuals for local ancestry analyses.** **(A)** Cross-validation scores from *K* = 3 to *K* = 10. **(B)** Admixture for *K = 5*.
(PDF)

**S27 Fig. Comparison of admixture and RFMix ancestry proportion estimates in DS2p and DS3p. (A-C) For DS2p:** Argentinean samples from the present study with reference panel that consists in 1KGP individuals from Africa, Europe and America [42] and Chilean individuals from [37]. Native American, European and African ancestry proportions estimates with RFMix vs with Admixture with *K = 7*. **(D-F) For DS3p:** Argentinean samples from [31] with reference panel that consists in 1KGP individuals from Africa, Europe and America [42]. Native American, European and African ancestry proportions estimates with RFMIx vs with

Admixture with *K = 5*.
(TIF)

**S28 Fig. Consistency of the masking procedure applied to DS2p and DS3p.** We compared the percentage of variants with same ancestry ditypes in DS2p and DS3p for American admixed individuals from the 1000 Genomes. Project.
(PDF)

**S1 Table. Sample information.** Sampling location, gender, uniparental lineages, Affymetrix QC metrics, color and point coding used for plots.
(XLS)

**S2 Table. Data sets (DS) analyzed throughout the article.**
(PDF)

**S3 Table. Ancestry proportion estimates in a worldwide context.** Ancestry proportion estimates from Admixture analyses with *K = 3* and *K = 8* at the worldwide level. The column names describe the labels attributed to each ancestry detecting for both Admixture analyses, as well as the hexadecimal code for the color used to represent it in the corresponding admixture plot. The columns "Point", "Color" and "cex" list the graphical parameters used to represent each individual in the different plots throughout the article.
(XLS)

**S4 Table. Native American cluster assignation.** Individual Native American cluster assignation is given for each of the three *K*-means procedures and for the consensus call (columns "F3", "PCA", "Admixture" and "Consensus"). The ancestry proportion estimates from Admixture analyses with *K = 3* on the masked data for Native American ancestry are also provided. The column names explicit the labels attributed to each ancestry detecting for both Admixture analyses as well as the hexadecimal code for the color. For admixed individuals (from the present study and [31]), the"Population" and"Region" columns list the locality and province, respectively, while for Native American population (from [37,38]) the"Population" and"Region" columns list the ethnic and main ethnic groups, respectively.
(XLS)

**S5 Table. *F*-statistics. (A)** $f_3$*(Target; S1, S2)* only for comparisons including Native American components **(B)** $f_4$*(YRI, Target; S1, S2)* only for comparisons including Native American components **(C)** $f_3$*(YRI; X, Y)* only for comparisons including Ancient Beringia, Mixe and Native American components **(D)** $f_3$*(YRI; X, Y)* where *X* and *Y* can be either an ancient group or one of the four Native American components. **(E)** $f_4$*(YRI, X; Ancient Beringia, Ancient)* only for comparisons including between a Native American components (*X*) and an ancient group (*Ancient*). **(F)** $f_4$*(YRI, Ancient, X, Y)* only for comparisons including two Native American components (*X, Y*) and an ancient group (*Ancient*). **(G)** $f_4$*(Ancient, X; Y, YRI)* only for comparisons including two Native American components (*X, Y*) and an ancient group (*Ancient*). YRI: Yoruba from 1KGP.
(XLSX)

## Acknowledgments

In memoriam of Raúl Carnese.

We thank Rolando González-José, Andrea Llera and Mariana Berenstein for the management and promotion of the Consorcio Poblar.

We thank all the participants of the different sampling campaigns across Argentina from which the newly genotyped samples are derived from.

We thank Ricardo A. Verdugo, Etienne Patin, Luca Pagani, Marta E. Alarcón Riquelme and María Teruel Artacho who kindly shared genotype data included in this study.

We thank Laura Fejerman for her proofreading of the manuscript.

## Author Contributions

**Conceptualization:** Pierre Luisi, Angelina García, Darío A. Demarchi, Hernán Dopazo.

**Data curation:** Pierre Luisi, Angelina García, Juan Manuel Berros, Josefina M. B. Motti, Darío A. Demarchi, Emma Alfaro, Eliana Aquilano, Carina Argüelles, Sergio Avena, Graciela Bailliet, Julieta Beltramo, Claudio M. Bravi, Mariela Cuello, Cristina Dejean, José Edgardo Dipierri, Laura S. Jurado Medina, José Luis Lanata, Marina Muzzio, María Laura Parolin, Maia Pauro, Paula B. Paz Sepúlveda, Daniela Rodríguez Golpe, María Rita Santos, Marisol Schwab, Natalia Silvero, Jeremias Zubrzycki, Virginia Ramallo.

**Formal analysis:** Pierre Luisi.

**Funding acquisition:** Pierre Luisi, Virginia Ramallo, Hernán Dopazo.

**Methodology:** Pierre Luisi, Juan Manuel Berros, Darío A. Demarchi, Hernán Dopazo.

**Project administration:** Pierre Luisi, Hernán Dopazo.

**Resources:** Juan Manuel Berros, Hernán Dopazo.

**Supervision:** Hernán Dopazo.

**Visualization:** Pierre Luisi.

**Writing – original draft:** Pierre Luisi, Angelina García, Hernán Dopazo.

**Writing – review & editing:** Josefina M. B. Motti, Darío A. Demarchi, Carina Argüelles, Sergio Avena, María Laura Parolin, Maia Pauro.

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
