## [Decision Letter · Decision Letter 0]

16 Mar 2020

PONE-D-20-02648

Fine-scale genomic analyses of admixed individuals reveal unrecognized genetic ancestry components in Argentina

PLOS ONE

Dear Dr Luisi,

Thank you for submitting your manuscript to PLOS ONE. After careful consideration, we feel that it has merit but does not fully meet PLOS ONE’s publication criteria as it currently stands. Therefore, we invite you to submit a revised version of the manuscript that addresses the points raised during the review process.

We would appreciate receiving your revised manuscript by Apr 30 2020 11:59PM. To enhance the reproducibility of your results, we recommend that if applicable you deposit your laboratory protocols in protocols.io, where a protocol can be assigned its own identifier (DOI) such that it can be cited independently in the future. For instructions see: http://journals.plos.org/plosone/s/submission-guidelines#loc-laboratory-protocols

We look forward to receiving your revised manuscript.

Kind regards,

Francesc Calafell

Academic Editor

PLOS ONE

Journal Requirements:

I have read the journal's policy and the authors of this manuscript have the following competing interests:

PL provides consulting services to myDNAmap S.A.

JMB is employed by Biocódices S.A.

HD is the scientific director of Biocódices S.A.

We note that one or more of the authors are employed by a commercial company: Biocodices S.A. and myDNAmap S.A.

4. We note that Figures 1, 2, 4 and S23 in your submission contain [map/satellite] images which may be copyrighted. All PLOS content is published under the Creative Commons Attribution License (CC BY 4.0), which means that the manuscript, images, and Supporting Information files will be freely available online, and any third party is permitted to access, download, copy, distribute, and use these materials in any way, even commercially, with proper attribution. For these reasons, we cannot publish previously copyrighted maps or satellite images created using proprietary data, such as Google software (Google Maps, Street View, and Earth). For more information, see our copyright guidelines: http://journals.plos.org/plosone/s/licenses-and-copyright.

1.    You may seek permission from the original copyright holder of Figures 1, 2, 4 and S23 to publish the content specifically under the CC BY 4.0 license. 

Reviewers' comments:

Reviewer's Responses to Questions

**Comments to the Author**

1. Is the manuscript technically sound, and do the data support the conclusions?

Reviewer #1: Yes

Reviewer #2: Yes

2. Has the statistical analysis been performed appropriately and rigorously? 

Reviewer #1: Yes

Reviewer #2: Yes

3. Have the authors made all data underlying the findings in their manuscript fully available?

Reviewer #1: Yes

Reviewer #2: Yes

4. Is the manuscript presented in an intelligible fashion and written in standard English?

Reviewer #1: Yes

Reviewer #2: No

5. Review Comments to the Author

Reviewer #1: The manuscript by Luisi et al is an important addition to our growing knowledge about the global and subcontinental ancestry of South American populations, focusing on Argentina. They used several up-to date methods to perform Population Genetics analyses and, we need to recognize the great effort compiling several publicly available genomic data from Native Americans, Africans and Europeans to be used as parental reference to their new genome-wide data of 87 individuals from Argentina. However, I have one major and some specific concerns, which should be addressed before publishing the manuscript in PLOS ONE:

MAJOR: The manuscript needs to be synthesized and more focused on the important results and on what the results mean rather than so many descriptions. The authors have very interesting results, but the reader gets distracted across the reading. Several descriptions of results and methodology could go to the Supplementary Material or Methods section.

Specific concerns and suggestions:

1) In the abstract, “then compared to different reference panels specifically built to run population structure analyses at a sub-continental level”, the word “run” should be changed to “perform” or other word of similar meaning.

2) In the Introduction, the authors gave a very good overview about previous studies using uniparental markers, but they should recognize recent important papers that used genome-wide data to study population structure in Central America (Moreno-Estrada et al 2013) and in Brazil (Kehdy et al. 2015). Specifically, about the African genetic components, there are some previous major studies that should be recognized (i.e. Mathias et al. 2016 and Gouveia et al. 2020). Also, did you look up at the African Voyages database (https://www.slavevoyages.org/) to see if they have information about the slave trade to Argentina?

3) At the beginning of “Results and Discussion”. It would help to have a subsection title “Studied Populations”

4) I think the authors should consider having one main figure describing the global population structure of Argentina (probably Figure 1). I know the authors focused on subcontinental ancestry; However, its important to have a sense of global population structure in Argentina without the need of access the supplementary material. For these global population structure, we have a clear correlation between PCA and ADMIXTURE analyses that should be integrated.

5) In the results, the authors constantly described the Peruvians and Chilean admixed samples before the results for Argentina, which distracted about the focus (Argentina results), which should be first.

6) The authors refer to the first main figure with ancestry results (Fig. 2) only in the middle of the manuscript. Again, the authors should begin with the important findings to avoid distracting the reader with descriptions.

7) It is not clear why the authors focused only on the Native American subcontinental components to perform F3 analyses. Also, the authors should highlight the importance of African and European subcontinental Ancestry in Argentina. Despite it agrees previous studies using uniparental data, this is the first time it is showed using genome-wide data, which we know that produces better estimates of ancestry proportions and subcontinental substructure when compared with uniparental data.

Reviewer #2: GENERAL COMMENTS

Luisi et al. have genotyped 94 DNA samples from various sites across Argentina using Axiom SNP arrays and integrated publicly available data from other studies to describe genetic affinities with modern and ancient South American populations, as well as other continental sources within Europe and Africa.

It is an important piece of work given the underrepresentation of South American populations in global genomic studies and for Argentina represents a pioneering effort towards a genomic characterization of its population. It has its limitations, mostly derived from the small sample size per subpopulation, but this seems to be compensated by re-analyzing previously published data from overlapping and surrounding populations in the region. This allowed them to assembly larger groups of samples per geographic region and to identify four major ancestry components in Argentina, one of them previously undescribed and putatively restricted to the Centralwest of Argentina (CWA).

In general it is methodologically sound and follows standard practices for the analysis of admixed genomes. My major comments will be oriented towards giving feedback about the general flow of the text with the intention of improving readability. There are many typos and minor (but important) grammatical errors throughout the text, so I suggest having a copyeditor to revise and polish the manuscript.

GENERAL COMMENTS

1. I found the abstract to be vague, long, and not appealing. It evokes claims of potential novelty without being clear about what exactly is new. If the novel finding is the identification of the CWA component of Native American origin, perhaps it should be highlighted more and earlier in the text.

2. The introduction is too long and overreaching. It has an ambitious scope but at the same time seems disproportionate when then the methodology is limited by the inferences from 87 individuals. I understand the motivation of being comprehensive in the introduction of concepts but perhaps the message can be more effectively conveyed by addressing more directly those elements that are missing in the literature and that these particular analyses will be contributing to.

3. What is the sampling scheme to enroll the 94 individuals in the study and to which extent it can be assumed their lineages are representative of the geographic regions they belong to? The family background is discussed further for a couple of samples that the authors acknowledge are possible cases of recent migration events within regions in Argentina, but I didn't easily find an overall description of the sampling criteria other than taking mt/Y haplogroups into account. This can be helpful to describe in an analogous way the POPRES samples are described for Europe.

4. The authors point to a couple of historical facts that could explain the affinity of Santiago and Cuyo samples to a novel component from Central Western Argentina. Although the historical accounts seem plausible, it will be better if the authors could formally test the specificity of such novel component, or at least suggest further analyses to do so. This is important given that many of the samples driving the signal of the Western Argentinean component are Chilean individuals from Santiago. Could this be reflecting a Huilliche-related component or any other unsampled population from the central southern cone?

5. Table S02 is very helpful and well organized. It's appreciated.

6. Fig S2: By excluding the Central and Northern Amerind samples from the unsupervised Admixture it is unclear whether the pink-salmon component assigned to CLM and MXL is the same as the one defined by the Chibchan-Paezan and Equatorial-Tucanoan groups, or if this model is missing such component. It would be helpful if the authors can clarify.

7. Fig 1. If samples are already grouped and colored by administrative region, what is the purpose of showing some sampling locations in different color shades within a given administrative region?

8. Fig 4. Consider inverting MDS values along the 1st dimension to better match the sampling locations in the map with the population clusters in MDS space. I think this will help interpretation of the figure.

9. Fig 4. Is the Lagoa Santa sample (Sumidouro) missing from the map in Fig4A?

SPECIFIC COMMENTS

p4. line 107: I think demography is not necessarily a "mechanism" in the context of shaping genetic diversity. I would suggest considering demographic "processes" instead.

p4. line 115: These kind of statements are just not informative - "We found that African origins in Argentina trace back from different regions". I would suggest rephrasing or removing.

p5. line 128: what do they mean by "broad scales" in South America, and thus what is the new "fine-scale" contributed by this paper? even if succinct, it should be stated upfront and spelled out so that the reader knows what is this about.

p5. line 129: better to say "we present *genomic* analyses"...

p5. line 131: typo, should say: limited number *of* markers...

p5. line 139: typo, should say: to *give* a general overview... (instead of "gives")

p7. line 178: the "16th century" and "the arrival of conquerors" seem to be two disconnected elements in this phrase: "has changed drastically from the 16th century, and the arrival of the first conquerors"...

are they meant to be disconnected or did the authors mean to say: "from the arrival of the first conquerors in the 16th century, until the 20th century"... ?

p8. line 190 & 191: there is no transition between these two sections. The topic changes abruptly from historical facts to giving examples of some of the recent genomic studies in the region. It would be better to walk the reader through and convey the message as to why such historical facts may have left a genetic signature in the modern Argentine population

p8. line 192: markers are not "through" the genome. I recommend finding a better wording in "thousands of autosomal markers through the genome for modern Argentine individuals"...

p8. line 206: to which novel component are the authors referring to in "described a Native American component not represented in the 1000 Genomes project"?

p8. line 209: it is not clear what the reader should understand by "achieving a fine-scale knowledge". What exactly is missing? and thus which gap is this paper helping to fill? I think those are key concepts the reader should be given upfront and clearly.

p12. line 294: I don't think the expected pattern is to be "striking" so there is no need to say so.

p13. line 308: it will be clearer if the authors can refer to the color used to identify each of the three components of Native American ancestry while describing Figure S2G (besides the CCP, CAN, and LWL acronyms).

p13. line 311. The authors might be quite familiar with the term Lowlands and the populations included in such groups but the reader might not. It will be clearer to spell this out the first time the term is introduced as well as in Fig S2. Is it supposed to include the Amazonian reference populations? the Argentinean too? and again, is this the same component observed in CLM and MXL? if that's the case it can be confusing to think about it as a "lowland-specific" component.

p14. line 327: I think it is expected that most Argentinean samples are assigned to all three Native American components if all the reference samples are from those sources. Local Native American samples from Argentina will be needed to determine whether that's the case or additional unsampled source populations are missing.

p15. line 356: by how much are the African and Native American ancestries underestimated? For clarity, it would be helpful to put some numbers to such comparison in the main text.

p16. line 376: it will be clearer if the authors can identify by color the "new component" observed in Iberians. Also, wouldn't the authors expect the Argentinean samples to exhibit higher proportions of the southern European component predominant in PROPES Italian samples (from Fig. S7C) given the great wave of European immigration from Italy in the 19th and 20th centuries?

p17. line 393: the proportion of African ancestry is rather low in the Argentinean dataset. Can the authors elaborate more on how this can compromise the resolution and reliability of their MDS analysis? did they set a minimum threshold of African ancestry to be included in the analysis?

p18. line 419: the legend of Fig 3 is extremely poor and a minimum of methods description should be given, together with some major highlights for the reader to focus on. Figure legends in general have very few details.

p20. line 468: check the use of his/her in the description of the Puerto Madryn individual's pedigree.

p21. line 495: I think it should say "By increasing the number of..."

p22. line 506: 32 individuals from where? One has to dive into Fig S15 to figure it out and if I'm not mistaken this 4th cluster is mostly accounted for CYA samples from Calingasta and Chilean samples from Santiago. The reader will appreciate if this is stated from the out set in the main text to ease reading flow. Also, is there any connection between the aDNA component detected at K=5 and the fourth cluster of ancestry represented by the 32 modern south American individuals? probably not, but given the current text flow they seem to be related...

p23. line 532: why not refer to the fourth component as CWA in "(iii) the fourth component and CCP exhibit higher genetic affinity between them"... and so on? I know the therm CWA is later introduced as a nomenclature suggestion for the new component, but the reading is just not easy if it takes too long to reveal it. Specially when the text is already referring to figures (and supplementary plots) where CWA is already labeled.

p23. line 552: what is the evidence for the maternal lineages of the Calingasta individuals being region-specific? presumably the authors are referring to mtDNA haplogroups restricted to or predominant in the Cuyo region?

p23. line 556: consider saying "in the Cuyo region".

p24. line 562: should read "analyses accounting *for* the genetic relationship..."

p24. line 574: important typo, should not say "pre-Colombian" but "pre-Columbian".

p25. line 578: did they mean "out of reach the expansion of"... ?

p25. line 586: consider this alternative phrasing - "the identification of an undescribed Native American component not previously reported from autosomal markers". I think that "never described" is unnecessarily categorical.

p25. line 589: consider saying "underrepresentation of *diverse* regions in Argentina" instead of "many".

p25. line 590: consider removing "specific". To say "Native American genetic diversity" is specific enough.

p27. line 621: while the authors rightly point out the underrepresentation of the LWL component in ancient studies, isn't it counterintuitive that the few ancient samples from Brazil do not show much affinity with the modern samples while all previous analyses do show Argentinean NEA and PPA samples clustering with LWL reference populations, stemming mostly from Amazonian indigenous groups across Brazil. It is noted that the authors argue for the extended spread of the LWL component across the region and the rather restricted geographic coverage of the ancient Brazilian samples, but it would be nice to see the authors elaborating a bit more on possible demographic scenarios underlying the lack of contribution of these ancient samples into modern populations assigned to the LWL component.

p27. line 633: couple of typos, it should say "where X is one *of* the four Native American *components*" (plural)

p27. line 634: The f3 vs age correlations are an interesting addition to the analyses. However, I didn't see a high level statement summarizing what's the main finding other than just saying that there are some correlations even after correcting for geography, etc. I think it will be helpful to offer the interpretation even if the authors feel is too obvious from the plots. It can be a simple phrase along lines like "We observed a significant relationship between X and Y (P = X), where the older the ancient sample the lower the shared drift with the modern components" or "where more recent ancient samples tend to show higher affinities with the modern components" (if that's the case).

Also, can the authors rule out that such correlations are not a function of DNA preservation and/or amount of good quality data from each of the ancient samples? One could think that the older samples show lower f3 values simply due to lower proportions of available data across the genome. Can the authors elaborate on that possibility? What is the missing data per aDNA sample in each of these correlations?

p28. line 658: in the figure legend I think you want to say "the symbol inside the square represents..." (instead of "the point within the square"). Also, avoid "the point code" and consider instead: "the legend for plotted symbols of ancient samples...".

p30. line 707: I get what the authors want to say, but such subtitle is just not reflecting the idea and it has poor grammatical flow. Please consider rewording.

p31. line 724: another typo, instead of "three Andes ancient groups" it should say "three *Andean* ancient groups". Also there should be a conjunction (maybe "of") between events and geographical expansions.

p32. line 742: I would suggest expanding the first sentence to something like "We studied genetic ancestry at the sub-continental scale in Argentinean individuals in the context of other South American populations".

p32. line 753: consider removing the word "from" three times

p33. line 768: It is not clear what the authors want to say with "These groups exhibit within population structure, and gene flow are most likely to have occurred among them after divergence." Please copyedit and consider rephrasing.

p33. line 771: should be "limit" instead of "limits".

p33. line 780: should be "pre-Columbian" instead of "pre-Colombian".

p34. line 790: consider this alternative phrasing:

This study is a joint effort of Argentinean institutions funded by the national scientific system, and represents the first milestone of the Consorcio Poblar, a national consortium for creating a public reference biobank to support biomedical genomic research in the Republic of Argentina [75]. Genomic knowledge of local populations should be a priority for developing countries to achieve an unbiased representation of diversity in public databases and the scientific development at a global scale.

6. PLOS authors have the option to publish the peer review history of their article (what does this mean?). If published, this will include your full peer review and any attached files.

Reviewer #1: No

Reviewer #2: Yes: Andres Moreno-Estrada

---

## [Author Response · Author response to Decision Letter 0]

7 May 2020

5. Review Comments to the Author

Reviewer #1: The manuscript by Luisi et al is an important addition to our growing knowledge about the global and subcontinental ancestry of South American populations, focusing on Argentina. They used several up-to date methods to perform Population Genetics analyses and, we need to recognize the great effort compiling several publicly available genomic data from Native Americans, Africans and Europeans to be used as parental reference to their new genome-wide data of 87 individuals from Argentina. However, I have one major and some specific concerns, which should be addressed before publishing the manuscript in PLOS ONE:

MAJOR: The manuscript needs to be synthesized and more focused on the important results and on what the results mean rather than so many descriptions. The authors have very interesting results, but the reader gets distracted across the reading. Several descriptions of results and methodology could go to the Supplementary Material or Methods section.

We understand this concern and addressed it. We reduced the descriptions of the results and some sanity check procedures have been moved to the Material and Methods section. The Results and Discussion section has been reduced ~1200 words (~20%) without losing its main content. We think that the Results/Discussion section is now much clearer and easier to follow for the reader. Thanks for the suggestions.

Specific concerns and suggestions:

1) In the abstract, “then compared to different reference panels specifically built to run population structure analyses at a sub-continental level”, the word “run” should be changed to “perform” or other word of similar meaning.

We now use “perform”.

2) In the Introduction, the authors gave a very good overview about previous studies using uniparental markers, but they should recognize recent important papers that used genome-wide data to study population structure in Central America (Moreno-Estrada et al 2013) and in Brazil (Kehdy et al. 2015). Specifically, about the African genetic components, there are some previous major studies that should be recognized (i.e. Mathias et al. 2016 and Gouveia et al. 2020). 

We now included citations to Khedy et al. 2015 and Gouveia et al. 2020. 

First, in the introduction (Line 171): “A previous study showed that Western and Central Western African ancestries are common across the Americas, particularly in Northern latitudes, while the influence of South/Eastern African ancestry is greater in South America [33]. In another study in Brazil, two African ancestries have been observed: a Western African one and another associated with Central East African and Bantu populations, the latter being more present in the Southeastern and Southern regions [34]”.

And then in Results and Discussion (Line 370): “In addition, the presence of Southeastern Africa maternal lineages in Argentina [27,28] is consistent with African ancestry of this origin identified in previous studies in other South America countries [33,34], and with the Eastern African ancestry identified here.”

We decided not to cite Mathias et al. 2016 since we do not find that their results are of direct interest for the scope of our study.

In order to keep the introduction focused on the subject of this article, we preferred not to cite literature concerning Central American. Note that Moreno-Estrada et al 2013 is cited later during the discussion (Line 362). 

Also, did you look up at the African Voyages database (https://www.slavevoyages.org/) to see if they have information about the slave trade to Argentina?

Thanks for recommending this very interesting database. We consulted it and we could not find particular information about the origins of the slaves in Argentina specifically. 

3) At the beginning of “Results and Discussion”. It would help to have a subsection title “Studied Populations”

Thanks for the suggestion. The new version includes such sub-section.

4) I think the authors should consider having one main figure describing the global population structure of Argentina (probably Figure 1). I know the authors focused on subcontinental ancestry; However, its important to have a sense of global population structure in Argentina without the need of access the supplementary material. 

Agreed. We now included Admixture analysis (with K=8) as Main Fig 2. This allows presenting earlier in the manuscript not only the three main continental ancestries but also of the main sub-continental ancestries discussed later.

For these global population structure, we have a clear correlation between PCA and ADMIXTURE analyses that should be integrated.

We included an additional Supplementary Figure for that and a mention in the main text (Line 263): “Moreover, the eigenvectors from PCA and the ancestry proportion estimates with Admixture are well correlated (S4 Fig).“

5) In the results, the authors constantly described the Peruvians and Chilean admixed samples before the results for Argentina, which distracted about the focus (Argentina results), which should be first.

Thanks. In the new version, we describe Argentinean samples first.

6) The authors refer to the first main figure with ancestry results (Fig. 2) only in the middle of the manuscript. Again, the authors should begin with the important findings to avoid distracting the reader with descriptions.

Agreed. This point has been addressed in Comment #4 from the same Referee.

7) It is not clear why the authors focused only on the Native American subcontinental components to perform F3 analyses. 

If the referee refers to using F-statistics to compare groups of Admixed individuals to reference groups of African or European individuals (either modern or ancient):

F-statistics analyses at the group level we performed for the Native American component aimed at addressing the question of a putative presence of an unrecognized Native American component that would explain why many individuals exhibit (i) mid proportion estimates for the three Native American ancestries represented in the reference panel, and (ii) intermediate positions in the PCA graph. Such question is not relevant for European and African ancestries.

If the referee refers to f3(YRI; Individual1, Individual2) as shown in S15 Fig to estimate genetic affinity between two individuals: In Europe and Africa, we did not used f3 to build inter-individual distances, and summarize the distance matrix with MDS as we did for America. 

First an MDS based on 1-f3(YRI; Individual1, Individual2) distances in Europa does not perform as well as. Indeed, the MDS from f3-based distances does not allow catching in detail the geographical cline in the European Reference Panel (see Figure below).

In Africa, we did not intent to run f3. The very low proportions of the genome with African ancestry in Admixed individuals imply a very reduced number of overlapping SNPs in masked data for each pair of Admixed individuals (few hundreds or thousands SNPs). Such reduced number of SNPs does not allow estimating informative f3 statistics. 

Moreover, we want to stress here that we did use f3 for pairwise individual comparisons in America only for Cluster assignation and not to describe the Native American ancestry composition in Argentina in the Main Text. For that purpose, we only relied on the results from Admixture and PCA. We wished to have the most robust assignation to a Native American Group as possible, leveraging different inter-individual distances. Since the results obtained with f3-based distances matched very well Admixture and PCA results, we decided to keep it for that purpose only. We thus could draw more robust boundaries among the different Native American Groups by removing few individuals with inconsistent assignation across the clustering method applied to three kinds of inter-individual distance matrix.

Also, the authors should highlight the importance of African and European subcontinental Ancestry in Argentina. Despite it agrees previous studies using uniparental data, this is the first time it is showed using genome-wide data, which we know that produces better estimates of ancestry proportions and subcontinental substructure when compared with uniparental data.

We agree that the results for European and African ancestry were a bit despised. We now put the PCA results for Europe as Main Figure. We also give more importance to the findings about European and African ancestries.

The last sentence of the Abstract (Line 95): “As for the European and African ancestries, we confirmed previous results about origins from Southern Europe, Western and Central Western Africa, and we provide evidences for the presence of Northern European and Eastern African ancestries.”

In Author summary (Line 106): “We confirmed that most of the European genetic ancestry comes from the South, although several individuals are related to Northern Europeans. We confirmed that the African origins in Argentina mainly trace back from Western and Central/Western regions, and we document some proportion of Eastern African origins poorly described before.”

In Conclusion (Line 623): “Our work shows that studying more admixed individuals, with a particular focus on extending the geographical coverage of the Argentinean territory, would help to identify the genetic legacy from secondary migration streams, and thus to get a better representation of the complex origins of African and European ancestries in the country.”

Reviewer #2: GENERAL COMMENTS

Luisi et al. have genotyped 94 DNA samples from various sites across Argentina using Axiom SNP arrays and integrated publicly available data from other studies to describe genetic affinities with modern and ancient South American populations, as well as other continental sources within Europe and Africa.

It is an important piece of work given the underrepresentation of South American populations in global genomic studies and for Argentina represents a pioneering effort towards a genomic characterization of its population. It has its limitations, mostly derived from the small sample size per subpopulation, but this seems to be compensated by re-analyzing previously published data from overlapping and surrounding populations in the region. This allowed them to assembly larger groups of samples per geographic region and to identify four major ancestry components in Argentina, one of them previously undescribed and putatively restricted to the Centralwest of Argentina (CWA).

In general it is methodologically sound and follows standard practices for the analysis of admixed genomes. My major comments will be oriented towards giving feedback about the general flow of the text with the intention of improving readability. There are many typos and minor (but important) grammatical errors throughout the text, so I suggest having a copyeditor to revise and polish the manuscript.

First of all, we apologize for the typos. Second, we are very grateful to the referee who clearly made an important effort to help us to improve the writing. 

We now asked a person speaking English fluently to perform a proofreading of the manuscript.

GENERAL COMMENTS

1. I found the abstract to be vague, long, and not appealing. It evokes claims of potential novelty without being clear about what exactly is new. If the novel finding is the identification of the CWA component of Native American origin, perhaps it should be highlighted more and earlier in the text.

We performed many changes in the abstract based on the Referee suggestions. We explicitly mention the novelty of the Northern European and Eastern African ancestries. We also mention the CWA component earlier in the abstract. Finally, we also reduced the number of words (by ~1/3) so it focuses more on the results and it appeals more potential readers.

2. The introduction is too long and overreaching. It has an ambitious scope but at the same time seems disproportionate when then the methodology is limited by the inferences from 87 individuals. I understand the motivation of being comprehensive in the introduction of concepts but perhaps the message can be more effectively conveyed by addressing more directly those elements that are missing in the literature and that these particular analyses will be contributing to.

Agreed. We drastically reduced the introduction (by a third) so it focuses more on the scope of the study. 

3. What is the sampling scheme to enroll the 94 individuals in the study and to which extent it can be assumed their lineages are representative of the geographic regions they belong to? The family background is discussed further for a couple of samples that the authors acknowledge are possible cases of recent migration events within regions in Argentina, but I didn't easily find an overall description of the sampling criteria other than taking mt/Y haplogroups into account. This can be helpful to describe in an analogous way the POPRES samples are described for Europe.

We did not consider describing the collections as in POPRES. Indeed, the samples were collected in 12 different sampling campaigns and describing each would not be particularly interesting, particularly because the data we present here will not probably be used as a reference panel like POPRES. However, we extended the description of the samples in Material and Methods, so the scheme to select the samples is more explicit.

Line 677: “We genotyped 94 individuals with the Axiom LAT1 array (Affymetrix) from 24 localities and 17 provinces across Argentina (Fig 1). These samples were selected among 240 collected by different population genetics groups (Consorcio PoblAR) during past sampling campaigns with a biological anthropology focus. According to the available information (e.g. interviews, genealogical information, etc.), each PoblAR research group selected for this study some samples, maximizing the odds that they come from individuals with greater Native American ancestry. For example, surnames were used as a proxy to achieve this objective and the permanence of ancestors in national territory has been another variable that was taken into account. The analyzed 94 samples were also selected to ensure an extended geographical range and were included when they presented sufficient DNA concentration and Native American maternal lineage. Moreover, among the males, we prioritized those with Native American paternal lineage.”

4. The authors point to a couple of historical facts that could explain the affinity of Santiago and Cuyo samples to a novel component from Central Western Argentina. Although the historical accounts seem plausible, it will be better if the authors could formally test the specificity of such novel component, or at least suggest further analyses to do so. This is important given that many of the samples driving the signal of the Western Argentinean component are Chilean individuals from Santiago. Could this be reflecting a Huilliche-related component or any other unsampled population from the central southern cone?

We agree that it is important to have the most extensive possible body of evidence concerning this new component. We added in Main Fig 7 (Relationship among the four Native American groups identified), the results for pairwise FST among groups. The important genetic differentiation between CWA and CCP is an extra evidence for the specificity of this novel component. We also demonstrate that removing Santiago individuals from CWA does not alter at all our results (S23 Fig). Moreover, as it is now more clearly stated in the revised manuscript, “all the Huilliche and Pehuenche individuals from Central Chile [37] have been consistently assigned to CCP” (Line 448). Altogether, we argue that CWA is not Huilliche-related despite the tight relationship between CWA and CCP since their early divergence, as suggested by the f-statistics we present in the manuscript.

We agree that CWA is a tentative name for this novel component, and further analyses are needed to confirm it. We stress this point in Conclusions (Line 653): “Further efforts are needed to better characterize the Native American ancestry component identified in the Central Western region of Argentina. Particularly, we encourage future studies to confirm the tentative geographical label that we suggest here, and to estimate its influence in the region.”

However, since there are very few Native American communities in the central region, addressing this question is not straightforward. We think that more ancient DNA samples from this region would help. This is now stressed more clearly in Results and Discussion (Line 613): “The archaeological record for which genetic data has been generated misrepresents CWA since its early divergence with CCP, as well as the common ancestors specific to these two components.”

5. Table S02 is very helpful and well organized. It's appreciated.

6. Fig S2: By excluding the Central and Northern Amerind samples from the unsupervised Admixture it is unclear whether the pink-salmon component assigned to CLM and MXL is the same as the one defined by the Chibchan-Paezan and Equatorial-Tucanoan groups, or if this model is missing such component. It would be helpful if the authors can clarify.

To answer this question, first we find necessary to clarify that we previously called the pink component Lowlands but we decided to change this label to Subtropical and Tropical Forests (STF). We discuss this point below addressing the referee comment concerning p13. line 31. 

We think that the use of this new label helps to dissipate the lack of clarity stressed by the referee. In our clustering approach for classifying individuals according to their main Native American Ancestry, we observe that the STF cluster encompasses populations not only from Gran Chaco and Amazonian populations but also from the North Andes (sampled almost exclusively in the Colombian territory). See the Native American groups plotted in pink in the South American map (Main Fig 5). Therefore, we think it is not surprising that CLM individuals exhibit a high proportion of the pink (STF) ancestry. However, we agree that it is a caveat to not explicitly discuss this point when presenting the Admixture results. We now discuss this (Line 274): “Such mixture pattern is not observed in other South American countries. Indeed, the Native American ancestry for Peruvian, Chilean and Colombian admixed samples is mainly represented by CAN, CCP and STF, respectively. This is consistent with the geographical area where the admixed individuals have been sampled, and the genetic ancestry of the indigenous communities from each country.”

Concerning MXL, we interpret that the high proportions of the pink ancestry could be due to back migrations into Central America previously described. We mention this possible scenario in the Introduction (Line 191): “In another genomic study of modern samples (in which the Southern Cone is only represented by the Gran Chaco region), it has been found that all non-Andean South American populations are likely to share a common lineage, while they are unlikely to share with the Andeans any common ancestor from Central America [39], supporting the hypothesis of many back migrations to Central America from non-Andean South American populations [39].” However, understanding past population movements between South America and Central America is still an open question. We would rather not distract the reader including such considerations in our manuscript for two reasons: (i) the relationship with Central America concerns mostly the Northern part of South America, and our work focuses on the Southern Cone, (ii) to explain the genetic affinity between non-Andean South American populations and Central American populations, the “back-migration into Central America” scenario is preferred with the current body of evidences. Altogether, we interpret that recent genetic influence of Central American populations on Argentina populations would be reduced.

Therefore, not having to address those questions that are beyond the scope of our study, we decided to now remove MXL (and PUR) individuals, and thus not to include Central and North indigenous communities, from our analyses. 

Summarizing our answer to this interesting point stressed by the referee: (i) we removed MXL and PUR from the analyses, (ii) we added a short text to discuss the pink ancestry in CLM, and (iii) we changed the label of this pink ancestry to Subtropical and Tropical Forests.

7. Fig 1. If samples are already grouped and colored by administrative region, what is the purpose of showing some sampling locations in different color shades within a given administrative region?

We think it is clearer to use broad groups of sampling location in our graphical representations. However, within a region, there are different provinces (internal lines in the map), which are actually the real political divisions in Argentina. Therefore, we used a color shade for each province, with similar color shades within the same region. We think that this is the most informative way to refer to the samples. Moreover, there is a technical limitation behind this decision: we wanted to use filled points for Argentinean samples so they are clearly highlighted in the plots, but in R there are only up to 5 different filled point types, that is less than the number of described locations in several regions.

8. Fig 4. Consider inverting MDS values along the 1st dimension to better match the sampling locations in the map with the population clusters in MDS space. I think this will help interpretation of the figure.

Thanks. We changed the MDS graphic for America (now Main Fig. 5).

9. Fig 4. Is the Lagoa Santa sample (Sumidouro) missing from the map in Fig4A?

Sumidouro was included in our analyses (4 out of 5 samples passed missing genotypes filtering): all the points that appear in the map and in the MDS are given in the legend insets. We acknowledge that it was really difficult to differentiate between overlapping points on the map, and we generated a new map avoiding this issue (now Main Fig. 5). 

SPECIFIC COMMENTS

p4. line 107: I think demography is not necessarily a "mechanism" in the context of shaping genetic diversity. I would suggest considering demographic "processes" instead.

This sentence has been removed

p4. line 115: These kind of statements are just not informative - "We found that African origins in Argentina trace back from different regions". I would suggest rephrasing or removing.

We agree with the referee and we reformulated this.

p5. line 128: what do they mean by "broad scales" in South America, and thus what is the new "fine-scale" contributed by this paper? even if succinct, it should be stated upfront and spelled out so that the reader knows what is this about.

This sentence had been removed and there is no reference to “broad-scale” in the article, what was indeed confusing. Now in the introduction, we think that it is clear that “fine-scale analyses” refer to the analysis of sub-continental level origins.

p5. line 129: better to say "we present *genomic* analyses"...

Thanks

p5. line 131: typo, should say: limited number *of* markers...

Typo has been fixed

p5. line 139: typo, should say: to *give* a general overview... (instead of "gives")

Typo has been fixed

p7. line 178: the "16th century" and "the arrival of conquerors" seem to be two disconnected elements in this phrase: "has changed drastically from the 16th century, and the arrival of the first conquerors"...

are they meant to be disconnected or did the authors mean to say: "from the arrival of the first conquerors in the 16th century, until the 20th century"... ?

We reformulated this sentence (Line 146): “As for the Native American component, it is difficult to study its origin focusing on present-day communities since their organization has changed drastically after the arrival of the first conquerors in the 16th century [21].”

p8. line 190 & 191: there is no transition between these two sections. The topic changes abruptly from historical facts to giving examples of some of the recent genomic studies in the region. It would be better to walk the reader through and convey the message as to why such historical facts may have left a genetic signature in the modern Argentine population

We now formulated a transition between the two sections (Line 155): “Due to the specificity of the Argentinean demographic history, a remaining challenge is to unravel which populations from each continent contributed to the genetic pool in nowadays Argentinean populations leveraging genotype data for hundreds of thousands of autosomal markers from the whole genome.”

p8. line 192: markers are not "through" the genome. I recommend finding a better wording in "thousands of autosomal markers through the genome for modern Argentine individuals"...

We reformulated the sentence.

p8. line 206: to which novel component are the authors referring to in "described a Native American component not represented in the 1000 Genomes project"?

We removed the reference to this result in the introduction since it was uninformative at this point.

p8. line 209: it is not clear what the reader should understand by "achieving a fine-scale knowledge". What exactly is missing? and thus which gap is this paper helping to fill? I think those are key concepts the reader should be given upfront and clearly.

We reformulated the sentence.

p12. line 294: I don't think the expected pattern is to be "striking" so there is no need to say so.

We removed “striking”.

p13. line 308: it will be clearer if the authors can refer to the color used to identify each of the three components of Native American ancestry while describing Figure S2G (besides the CCP, CAN, and LWL acronyms).

We added it, not only for this section of the text, but also for every description of Admixture plots.

p13. line 311. The authors might be quite familiar with the term Lowlands and the populations included in such groups but the reader might not. It will be clearer to spell this out the first time the term is introduced as well as in Fig S2. Is it supposed to include the Amazonian reference populations? the Argentinean too? and again, is this the same component observed in CLM and MXL? if that's the case it can be confusing to think about it as a "lowland-specific" component.

Before the first submission, we had extensive discussions about the label we could attribute to the pink component. It was not an easy decision due to the wide geographical range of the reference Native American individuals it encompasses: Gran Chaco, Amazonian and North Andes. The later is actually not Lowlands, but at the time of the first submission, we thought that Lowlands was the best option among the ones we could think about. After the referee comment, we discussed again this point and the alternative name “Subtropical and Tropical Forests” has been suggested. We think it is the best name we can assign to this pink component since it matches perfectly the geographical range of the indigenous communities in which this component prevails, what avoids potential confusion.

p14. line 327: I think it is expected that most Argentinean samples are assigned to all three Native American components if all the reference samples are from those sources. Local Native American samples from Argentina will be needed to determine whether that's the case or additional unsampled source populations are missing.

For clarity of the manuscript, we think it is better not discuss in details the ancestry composition at the sub-continental level when presenting the results in a worldwide context. This is why we do not mention the possibility of an unsampled source population. 

We now give more emphasis on formulating the two competing scenarios to explain this pattern in the section relative to Native American ancestry (Line 415) “Many individuals from the Cuyo and Pampean regions of Argentina (San Juan and Córdoba provinces as well as South of Buenos Aires province) exhibit intermediate position in PCA (Fig 5) and mid proportion estimates with Admixture (Fig 6). This pattern can be interpreted as the result of a mixture between different ancestries (scenario 1) or relative limited shared history with any of them (scenario 2).“

We focused most of the study of Native American specific ancestry to comparisons with ancient samples to contrast whether the pattern in Argentina of mixed Native American components is really due to a mixture between those components or because Native American references are missing. We think that our work gives robust evidences for the latter.

Moreover, we wish to highlight that a main point of our article is that we argue that “getting Native American samples from Argentina” representing all the Native American genetic ancestry is a hard, if possible at all, task since very few communities remain nowadays. The communities are mostly distributed in the extreme sides of the national territory. In the article, we stressed this point several times, for example:

Introduction (Line 146): “As for the Native American component, it is difficult to study its origin focusing on present-day communities since their organization has changed drastically after the arrival of the first conquerors in the 16th century [21].“

Conclusion (Line 636): “Having identified this component from admixed individuals demonstrates that focusing only on indigenous communities is insufficient, at least in Argentina, to fully characterize the Native American genetic diversity and decipher the pre-Columbian history of Native Americans. Indeed, most indigenous communities have been culturally annihilated and invisibilized [68,69] to the point that several Argentinean regions were considered “Indian free” in the mid-20st century [70]. However, the cultural incorporation did not necessarily imply a biological extinction. Although studies based on samples from indigenous communities [37–39] provide decisive information to understand the evolutionary history of Native American ancestry, alternative strategies must be considered to fill this gap in the effort to more fully describe the Native American ancestry (e.g. see [71]).“

We hope that we have clarified our position about the sampling scheme only focusing on indigenous communities in the restructured version of the manuscript thanks to the Referees’ suggestions.

p15. line 356: by how much are the African and Native American ancestries underestimated? For clarity, it would be helpful to put some numbers to such comparison in the main text.

Following Referee #1 suggestion to keep the Results section focused on the main results, this part moved to Material and Methods (Line 777). S27 Fig (S5 Fig in previous version) aims at answering this point. However, redundant information in the previous version of this figure made its visualization rather discouraging. We simplified the figure and added boxplots as insets so that it is easier to quantity the differences of ancestry proportions estimates with RFMIX and Admixture. 

Moreover, we performed an additional quality control for the masking scheme (this is mentioned in the text, Line 779 and S28 Fig): we checked how consistent are the masking in 1000 Genomes admixed individuals for the ~170K SNPs present both in DS2p and DS3p (the two datasets for which RFMIX has been run separately). We found that 95% Admixed 1000 Genomes individuals have more than 99% such SNPs with the same masking in DS2p and DS3p while the individual with the lower consistency rate had only 4% SNPs with inconsistent masking 

p16. line 376: it will be clearer if the authors can identify by color the "new component" observed in Iberians. 

Done.

Also, wouldn't the authors expect the Argentinean samples to exhibit higher proportions of the southern European component predominant in PROPES Italian samples (from Fig. S7C) given the great wave of European immigration from Italy in the 19th and 20th centuries?

We now address this question in the manuscript. We performed new European Ancestry Specific analyses including CLM from 1000 Genomes (no PEL individual with masked genotype data passed the filtering step for missing genotype rate). Line 314: “Samples from Argentina and Chile exhibit higher proportions of Southeastern/Italian and Northern European ancestries than Colombians, as well as lower Iberian ancestry proportions (S8 Fig). We observed no significant difference in the proportions of any European ancestry between Argentinean and Chilean samples (S8 Fig). However, both PCA and Admixture shows that the individuals with most Southeastern/Italian ancestry are from Argentina, This is consistent with a previous study [31], and it can be explained by the many arrivals from Italy during the great wave of European immigration in the 19th and 20th centuries [22]. “

p17. line 393: the proportion of African ancestry is rather low in the Argentinean ataset. Can the authors elaborate more on how this can compromise the resolution and reliability of their MDS analysis? did they set a minimum threshold of African ancestry to be included in the analysis?

We acknowledge that we applied a rather loose criteria since we kept individuals with more than 5% SNPs with African ancestry ditypes assigned (--mind 0.95 flag with plink). But we think that this does not affect our conclusions.

We now added a sentence to briefly discuss this point (Line 350): “Although, the important missing genotype rate in masked data for admixed individuals could bias PCA and Admixture results, the results obtained by both methods are highly consistent for admixed individuals (S11 Fig).”

In what follows, we provide more elements.

Because of missing genotypes we expect that samples with masked genotypes would tend to:

- Have values close to 0 for any PC with PCA (although the lsq transformation implemented in smartpca tends to reduce this bias)

- Have overestimated ancestry proportions with Admixture for the ancestries most represented in the reference panel (in our case: Bantu-influenced and Western African ancestries). 

If we have a look at individual PCs (now shown as S9 Fig), we observe:

- Argentinean individuals have negative 0 for PC1, and are grouped with Bantu Eastern and Western African. That is they are not evenly distributed around 0 as expected if their value was only explained by the fact that they have many missing data. This demonstrates that Argentineans have actually most of their African Ancestry explained by these three ancestries.

- A clear gradient for Argentineans on PC4, between Western African (top) and Bantu-influenced populations (bottom). 

- Some Argentinean individuals (from Chepes, Formosa and Tucuman) have positive values on PC3 that discriminates Eastern Africans to the other African populations. 

- The same individuals have the greatest Eastern African ancestry proportions with Admixture (in orange). This ancestry is the most poorly represented in the African reference panel we built. 

The very important consistency for PCA and Admixture analyses for Admixed individuals is an important argument against a potential bias due to missing genotype. Moreover, our conclusions match historical records and maternal lineage analyses (citation in the manuscript Line 369-373). Altogether, we are convinced that any artifact due to missing genotypes does not drive our conclusions. Most of the African ancestry is really explained by Western African and Bantu-influenced ancestries. The Eastern African ancestry we observe for some Argentinean individuals (the novel result) is really present in Argentina.

Note: after addressing the referee’s comment, we decide to present the African ancestry specific PCA in a different way (individuals PC graphs instead of the MDS based on distance from the 5 first PCs as previously showed). We implemented this change and adapted the manuscript section accordingly.

p18. line 419: the legend of Fig 3 is extremely poor and a minimum of methods description should be given, together with some major highlights for the reader to focus on. Figure legends in general have very few details.

We agree that the legends of figures ancestry-specific PCA and Admixture analyses were not including enough details. We improved them (Main Figs 2-6)

p20. line 468: check the use of his/her in the description of the Puerto Madryn individual's pedigree.

This sentence has been removed.

p21. line 495: I think it should say "By increasing the number of..."

Corrected.

p22. line 506: 32 individuals from where? One has to dive into Fig S15 to figure it out and if I'm not mistaken this 4th cluster is mostly accounted for CYA samples from Calingasta and Chilean samples from Santiago. The reader will appreciate if this is stated from the out set in the main text to ease reading flow. Also, is there any connection between the aDNA component detected at K=5 and the fourth cluster of ancestry represented by the 32 modern south American individuals? probably not, but given the current text flow they seem to be related…

We acknowledge that the reading was rather difficult. Based on the reviewer’s very useful suggestions, we changed the description of the results. We believe that in the new version of the manuscript, it is now clear from the out set in the main text that the 4th cluster is mostly accounted for Cuyo samples from Calingasta and Chilean samples from Santiago. 

We also acknowledge that the description of admixture results with K=4 and K=5 was misleading for the reader. We reformulated the entire section so it articulates better what follows of the manuscript and the description of the results (and their limitations) from admixture and PCA (Lines 415-430). 

p23. line 532: why not refer to the fourth component as CWA in "(iii) the fourth component and CCP exhibit higher genetic affinity between them"... and so on? I know the therm CWA is later introduced as a nomenclature suggestion for the new component, but the reading is just not easy if it takes too long to reveal it. Specially when the text is already referring to figures (and supplementary plots) where CWA is already labeled.

Before the first submission, we hesitated a lot to name directly the fourth component CWA. We now have restructured this section following his suggestion. 

p23. line 552: what is the evidence for the maternal lineages of the Calingasta individuals being region-specific? presumably the authors are referring to mtDNA haplogroups restricted to or predominant in the Cuyo region?

We changed the sentence (Line 445): “The genealogical record for the Calingasta individuals attests to a local origin of their direct ancestors up to two generations ago, and they have mtDNA sub-haplogroups predominant in the Cuyo region (S1 Table; [56,57]).” The reader can find more precision in the cited works.

p23. line 556: consider saying "in the Cuyo region".

Done.

p24. line 562: should read "analyses accounting *for* the genetic relationship..."

Done.

p24. line 574: important typo, should not say "pre-Colombian" but "pre-Columbian".

We fixed the typo

p25. line 578: did they mean "out of reach the expansion of"... ?

Thanks

p25. line 586: consider this alternative phrasing - "the identification of an undescribed Native American component not previously reported from autosomal markers". I think that "never described" is unnecessarily categorical.

Thanks. We changed it.

p25. line 589: consider saying "underrepresentation of *diverse* regions in Argentina" instead of "many".

Thanks. We changed it.

p25. line 590: consider removing "specific". To say "Native American genetic diversity" is specific enough.

Thanks. We changed it.

p27. line 621: while the authors rightly point out the underrepresentation of the LWL component in ancient studies, isn't it counterintuitive that the few ancient samples from Brazil do not show much affinity with the modern samples while all previous analyses do show Argentinean NEA and PPA samples clustering with LWL reference populations, stemming mostly from Amazonian indigenous groups across Brazil. It is noted that the authors argue for the extended spread of the LWL component across the region and the rather restricted geographic coverage of the ancient Brazilian samples, but it would be nice to see the authors elaborating a bit more on possible demographic scenarios underlying the lack of contribution of these ancient samples into modern populations assigned to the LWL component.

We reformulated this part, so we discuss different potential (perhaps complementary) reasons to explain the lack of contribution of these ancient samples into modern STF (Line 527): “The fact that there is no ancient sample group exhibiting outstanding genetic affinity with STF points to the underrepresentation of this component in ancient samples. First, the geographical range covered by ancient samples that could represent this component is restricted to Brazil, while STF is a heterogeneous group that includes relatively isolated populations [39] from the Gran Chaco, the Amazonas and Northern Andes. Moreover, the most recent samples that could represent STF are aged ~6700BP, and gene flow with other components since then may have contributed to dissolve the genetic affinity of STF with ancient samples in Brazil analyzed here.”

We provide here more details about this discussion. Posth et al. 2018 described a population replacement in Brazil after 9600BP (the age of Lapa Do Santo sample). It would mean that the second ancestry stream that has potentially remained in the region since then is only represented by Laranjal ~ 6700BP. Since 6700BP, gene flow between the STF and other components must have dissipated the genetic affinity of STF component with this ancient group. We provide here some analyses F-statistics that demonstrate that (to the limit of the statistical resolution due to reduce number of SNPs when including two ancient DNA group in the comparison) 

1. STF has higher genetic affinity with Laranjal 6700BP than with the older ancient DNA samples from Brazil and Belize. Even if the trend is not outstanding, it is consistent with the population replacement suggested by Posth et al. 

2. STF has higher genetic affinity with Laranjal 6700BP than with the oldest ancient DNA samples (from both the Southern Cone and Central Andes), consistent with the fact that STF is related to ancient populations from Brazil. 

Looking closer (we removed the historical Kaweskar ~100B in what follows): there is a clear correlation between the age of the Ancient Sample and the f4 in both the Andes and Southern Cone. That is the genetic affinity of STF to ancient samples from other regions (relative to Laranjal 6700BP) increases with time. Consistent with gene flow between STF and ancient populations represented by Andes aDNA samples and between STF and the ancient populations represented by Southern Cone aDNA samples.

p27. line 633: couple of typos, it should say "where X is one *of* the four Native American *components*" (plural)

We fixed the typo.

p27. line 634: The f3 vs age correlations are an interesting addition to the analyses. However, I didn't see a high level statement summarizing what's the main finding other than just saying that there are some correlations even after correcting for geography, etc. I think it will be helpful to offer the interpretation even if the authors feel is too obvious from the plots. It can be a simple phrase along lines like "We observed a significant relationship between X and Y (P = X), where the older the ancient sample the lower the shared drift with the modern components" or "where more recent ancient samples tend to show higher affinities with the modern components" (if that's the case).

Thanks for the suggestion. We changed the text (Line 538): “We observed a statistically significant relationship between the age of the ancient Southern Cone samples and their genetic affinity with CCP and CWA. This means that the older the ancient sample from the Southern Cone, the lower the shared drift with CCP and CWA.”

Also, can the authors rule out that such correlations are not a function of DNA preservation and/or amount of good quality data from each of the ancient samples? One could think that the older samples show lower f3 values simply due to lower proportions of available data across the genome. Can the authors elaborate on that possibility? What is the missing data per aDNA sample in each of these correlations?

The number of SNPs for all the f-statistics estimated in the manuscript is given in S5 Table.

We agree that due to DNA damage, the number of SNPs with genotype data tends to decrease with the age of the ancient samples, what could induce a bias towards significant positive correlations between the age of ancient samples and their genetic affinity with modern Native American components. We discarded this possibility because such bias would imply similar correlations for the fourth components, a pattern we did not observe. In addition, we did not observe significant correlations between the number of SNPs to estimate f3 or f4 and ancient sample age for either the Andes or the Southern Cone (P-value for Spearman correlation test > 0.25).

However, in order to dissipate any doubt on this potential confounding factor, we addressed this question including an extra correction in our analysis that did not change the trends identified (Line 543). “These patterns could be due to a relationship between geography and the age of the ancient samples because the most recent samples are concentrated in the Southern tip of the subcontinent (Fig 5A). Moreover, the number of SNPs with genotype data tends to decrease with the age of the ancient samples due to DNA damage, and thus inducing a potential bias towards significant positive correlations. To simultaneity correct both these two putative confounding effects, we repeated the analyses but using a correction for the ancient sample age (the residuals of the linear regression between the age of the ancient samples and their geographic coordinates) and a correction for genetic affinity estimates (the residuals of the linear regression between f3 and the number of SNPs to estimate it). This correction intensified the relationship described for CCP and CWA (Fig 8C). It also allowed to actually identifying significant relationships for STF and CAN. On the other hand, CAN is the only modern Native American component that exhibits a significant relationship between its genetic affinity with ancient Andean samples and their age (Fig 8B). This pattern holds after correction for geography (Fig 8D)”

p28. line 658: in the figure legend I think you want to say "the symbol inside the square represents..." (instead of "the point within the square"). Also, avoid "the point code" and consider instead: "the legend for plotted symbols of ancient samples...".

Thanks. We included these suggestions

p30. line 707: I get what the authors want to say, but such subtitle is just not reflecting the idea and it has poor grammatical flow. Please consider rewording.

We removed the subtitle that was indeed poorly informative. The results described after that subtitle are totally articulated now with the previous section into a unique one now called “Early divergence among the four Native American components in Argentina”

We also reformulated the sentence pointed by the referee (Line 602). “In order to get insights into the past genetic influence among the four components since their divergence, we applied a last f4-statistics analysis (S5G Table; S22 Fig).”

p31. line 724: another typo, instead of "three Andes ancient groups" it should say "three *Andean* ancient groups". Also there should be a conjunction (maybe "of") between events and geographical expansions.

This sentence has been removed.

p32. line 742: I would suggest expanding the first sentence to something like "We studied genetic ancestry at the sub-continental scale in Argentinean individuals in the context of other South American populations".

Thanks. Done.

p32. line 753: consider removing the word "from" three times

Done.

p33. line 768: It is not clear what the authors want to say with "These groups exhibit within population structure, and gene flow are most likely to have occurred among them after divergence." Please copyedit and consider rephrasing.

We agree that this sentence was not clear, and we removed it. We added the idea of within group genetic structure in the previous sentence. We removed the reference to gene flow since it was not relevant for the idea developed here. 

Line 647: “Studying admixed individuals can be complex, and leveraging a pure statistical approach, we grouped individuals from rather culturally, ethnically, linguistically, and genetically heterogeneous groups to represent the four Native American components discussed here.”

p33. line 771: should be "limit" instead of "limits".

We fixed the typo.

p33. line 780: should be "pre-Columbian" instead of "pre-Colombian".

We fixed the typo.

p34. line 790: consider this alternative phrasing:

This study is a joint effort of Argentinean institutions funded by the national scientific system, and represents the first milestone of the Consorcio Poblar, a national consortium for creating a public reference biobank to support biomedical genomic research in the Republic of Argentina [75]. Genomic knowledge of local populations should be a priority for developing countries to achieve an unbiased representation of diversity in public databases and the scientific development at a global scale. 

 Thanks for the suggestion. We included this phrasing

---

## [Editor Report · Decision Letter 1]

13 May 2020

Fine-scale genomic analyses of admixed individuals reveal unrecognized genetic ancestry components in Argentina

PONE-D-20-02648R1

Dear Dr. Luisi,

We are pleased to inform you that your manuscript has been judged scientifically suitable for publication and will be formally accepted for publication once it complies with all outstanding technical requirements.

With kind regards,

Francesc Calafell

Academic Editor

PLOS ONE
---

## [Editor Report · Acceptance letter]

26 Jun 2020

PONE-D-20-02648R1 

Fine-scale genomic analyses of admixed individuals reveal unrecognized genetic ancestry components in Argentina 

Dear Dr. Luisi:

I'm pleased to inform you that your manuscript has been deemed suitable for publication in PLOS ONE. Congratulations! Your manuscript is now with our production department. 

Kind regards, 

on behalf of

Dr. Francesc Calafell 

Academic Editor

PLOS ONE